# Eliciting local knowledge of ecosystem services using participatory mapping and Photovoice: A case study of Tun Mustapha Park, Malaysia

**Voon-Ching Lim**[1]*, **Eva Vivian Justine**[2,3], **Kamila Yusof**[2], **Wan Nur Syazana Wan Mohamad Ariffin**[2], **Hong Ching Goh**[2,4], **Kamal Solhaimi Fadzil**[5,6]

1 School of Science, Monash University Malaysia, Bandar Sunway, Selangor, Malaysia, 2 Department of Urban and Regional Planning, Faculty of Built Environment, Universiti Malaya, Kuala Lumpur, Malaysia, 3 Kudat Turtle Conservation Society, Kudat, Sabah, Malaysia, 4 Centre For Sustainable Urban Planning & Real Estate (SUPRE), Faculty of Built Environment, Universiti Malaya, Kuala Lumpur, Malaysia, 5 Department of Anthropology and Sociology, Faculty of Arts and Social Sciences, Universiti Malaya, Kuala Lumpur, Malaysia, 6 Centre for Malaysian Indigenous Studies, Universiti Malaya, Kuala Lumpur, Malaysia

* Lim.VoonChing@monash.edu

**Data Availability Statement:** All relevant data are within the paper and its Supporting Information files.

## Abstract

Protected areas in Malaysia have always been managed using top-down approach that often exclude the local communities, who are the main users of ecosystem services, from the planning and management. However, a newly established multiple-use marine park in Malaysia, Tun Mustapha Park (TMP), aims for inclusivity in managing the park. This research explores different participatory approaches (i.e. participatory mapping and Photovoice) to understand the ecosystem services and the dynamics surrounding the services in TMP. Community-based organisations and a mariculture farm in TMP were invited to participate in this work. The participants mapped the ecosystem services and provided in-depth qualitative data that supported the maps, besides highlighting ecological, sociocultural and economic issues surrounding the ecosystem services. Furthermore, the participants provided suggestions and recommendations that carry political effects. Therefore, the participatory approaches employed here had provided rich visual and spatial data to enhance the ecosystem-based management of TMP besides empowering the participants to voice out for their communities. The results generated from this work were also further utilised to fill in the gaps of knowledge in a separate ecosystem service assessment matrix. However, the output from participatory approaches should not be considered as the ultimate outcome but rather as supplement to the planning and management of TMP due to potential human errors and biases. Although the participatory approaches came with limitations and challenges that may have affected the findings here, these nonetheless had provided support to the capability of local communities to provide information crucial for management of protected areas as well as room for improvement for further work.

**Funding:** This project was supported by The Rufford Foundation in the form of Rufford Small Grant awarded to VCL [reference no. 28095-1] (this grant is hosted by the Universiti Malaya under reference no. IF050-2019) and by the United Kingdom Research and Innovation (UKRI) through Global Challenges Research Fund (GCRF) awarded to the Blue Communities Malaysian Case Study via HCG [under grant agreement reference no. NE/P021107/1] (this grant is hosted by the Universiti Malaya under reference no. IF052-2017).

**Competing interests:** The authors declare that they do not have competing financial interests or personal relationships that could influence the work reported here.

## Introduction

Ecosystem services (ES) are the benefits that human obtain directly and indirectly from the ecosystems which contribute to their well-being [1–3]. Millennium Ecosystem Assessment [2] and TEEB [3] categorised ES into provisioning services (e.g. food and medicine), cultural services (e.g. recreation, spiritual experience and educational opportunities), regulating services (e.g. pollination and carbon storage), and supporting services (e.g. production of oxygen and habitats for species). The increasing reliance of human on ES to improve the living standard is fuelled by the rapid global population growth, thus creating an imbalance between social economic development and supply of ES [4, 5]. Unsustainable use of ES and the consequent modifications to ecosystems could result in adverse impacts such as population decline of food species and climate change, which in turn affects human well-being especially those who utilise ES as main source of livelihood [2].

The primary purpose of a protected area is to conserve the biodiversity within its boundaries, though many policymakers and authorities are increasingly emphasising ES in the area to balance the trade-offs between conservation and natural resource use in order to meet ecological and social needs [6]. When developing an integrated management plan for a protected area, understanding the supply and demand of ES in within is crucial [7]. ES supply is the capability of an area to offer ecosystem goods and services within a time period, whereas ES demand is the ecosystem goods and services being consumed and used in the area over a time period [8]. High level of demand exceeding level of supply in an area may lead to unsustainable use of ecosystem goods and services. Therefore, understanding the drivers of ES, identifying mismatches between ES supply and demand, and mapping the ES within the area could facilitate the evaluation of trade-offs and synergies associated with decision-making and management for natural resource use [9–11]. Recognising the local use of ES could facilitate the implementation of appropriate strategies which can maintain the quality of the ecosystem without compromising the traditional benign use of natural resources [12, 13]. Furthermore, this could minimise the conflicts among stakeholders by ensuring a compromise between the local communities' needs and the authorities' conservation effort [6].

The conventional top-down approach in managing terrestrial and marine protected areas often excludes local communities who depend on the ES within the area from decision-making processes [1, 13]. Decision-making processes should be inclusive to gain support from all stakeholders as the establishment and management of protected areas can affect different groups of stakeholders who have different interests, needs and experience [13, 14]. Including local communities in the decision-making processes could provide insights into the ecosystems in the protected area and how the ES are being utilised particularly as source of food and income [15–17]. Such local knowledge include sustainable traditional use of ES that often have minimal adverse impact on the environment as well as conflicts pertaining to the use of ES due to the differing interests between stakeholders [13, 18]. Moreover, cultural ES which shapes the local and spiritual beliefs, cultural heritage, and traditional skills often require input from local communities [19]. Therefore, incorporation of local knowledge in decision-making could enhance the management of protected area besides empowering the local communities to conserve their homeland [20, 21].

One way to encourage the application of local knowledge in conservation is through participatory research, where local communities are involved in collecting, providing and/or processesing data voluntarily in scientific enquiry and ultimately in decision-making [19, 22]. With appropriate protocols and training, local communities can collect data of good quality for scientific research [16, 23]. Two approaches that have been commonly used in participatory research to assess the ES within an area are (i) participatory mapping [6, 12, 24, 25] and (ii)

Photovoice [1, 26–28]. Participatory mapping allows participants to map ES within an area, locate the conflicts and synergies between the ES and other land use, and highlight areas where particular ES are being threatened [29]. Photovoice provide opportunity for participants to highlight local issues in self-explanatory images that are unknown to outsiders anonymously, allowing participants to photograph freely [26]. When mapping and photography are conducted together with group discussion, participants could highlight issues and solutions with consensus view [12] besides allowing participants to provide a more complete picture of issues within their residence [27].

Studies from Thailand [27], Kenya [1] and Uruguay [21] have demonstrated how participatory research provided rich qualitative local knowledge that further supports the management of marine protected areas. Furthermore, participatory research can be an educational tool that instils pride and interest in the communities to protect their homeland [30]. This is particularly important for newly established and large-sized protected areas that are governed by small management team with limited physical capacity for implementing action plans such as the case in Tun Mustapha Park, Malaysia (TMP). Given the potential of participatory research to support management of marine protected areas, we employed participatory mapping and Photovoice to better understand the marine-associated habitats and ES in TMP. Through participatory mapping, we attempted to determine where and how the marine habitats are utilised by the communities, and the anthropogenic impacts on the habitats based on local knowledge and perception. Using Photovoice, we aimed to explore the aspects of coastal communities' life that are associated with the marine ecosystem. These participatory approaches could allow the communities to further share their knowledge, experiences and certain aspects of their life that are associated with the ES. We also discussed our experience of using the participatory approaches, which may be helpful to future work besides highlighting the potential of local knowledge in spatial mapping and ecosystem-based management of marine ecosystems.

## Method

### Study site

Tun Mustapha Park (TMP) is located at northern Sabah, along the boundary between Malaysia and Philippines [31, 32] (Fig 1). It is currently one of the largest marine parks in Malaysia, covering an area of 8987 $km^2$ at the north coast of the Sabah state and includes the Kudat-Banggi Priority Conservation Area in Sulu-Sulawesi Marine Ecoregion. The marine ecosystem in TMP serves as a source of seafood and livelihood for more than 85,000 coastal inhabitants with diverse ethnic groups [31, 32]. Yet TMP is threatened by destructive fishing activities such as fish bombing and cyanide fishing which causes damage to the coral reefs and seagrasses within the park [31–33]. Land clearing and coastal development also resulted in the loss of mangrove forests and consequently the loss of nursery grounds for many seafood species, which in return impact the livelihood and food security of the local communities [32, 34].

Such threats were the drivers of the establishment of TMP in 2016 which is divided into four zones; (i) no-take zone where extractive activities are prohibited, (ii) community-use zone where non-destructive small-scale and traditional fishing activities are allowed and nearby communities could take part in managing the natural resources, (ii) multiple-use zone for low impact activities including non-destructive and small-scale fishing activities as well as sustainable development activities (e.g. tourism and recreation) are allowed, and (iv) commercial fishing zone where large-scale extractive fishing practices are allowed [31, 35]. TMP is unique as it is the first multiple-use marine protected area in Malaysia, though only underwater habitats were formally gazetted within TMP at the time of writing, which falls under the jurisdiction of Sabah Parks under the Parks Enactment 1984 [35]. The intertidal mangroves

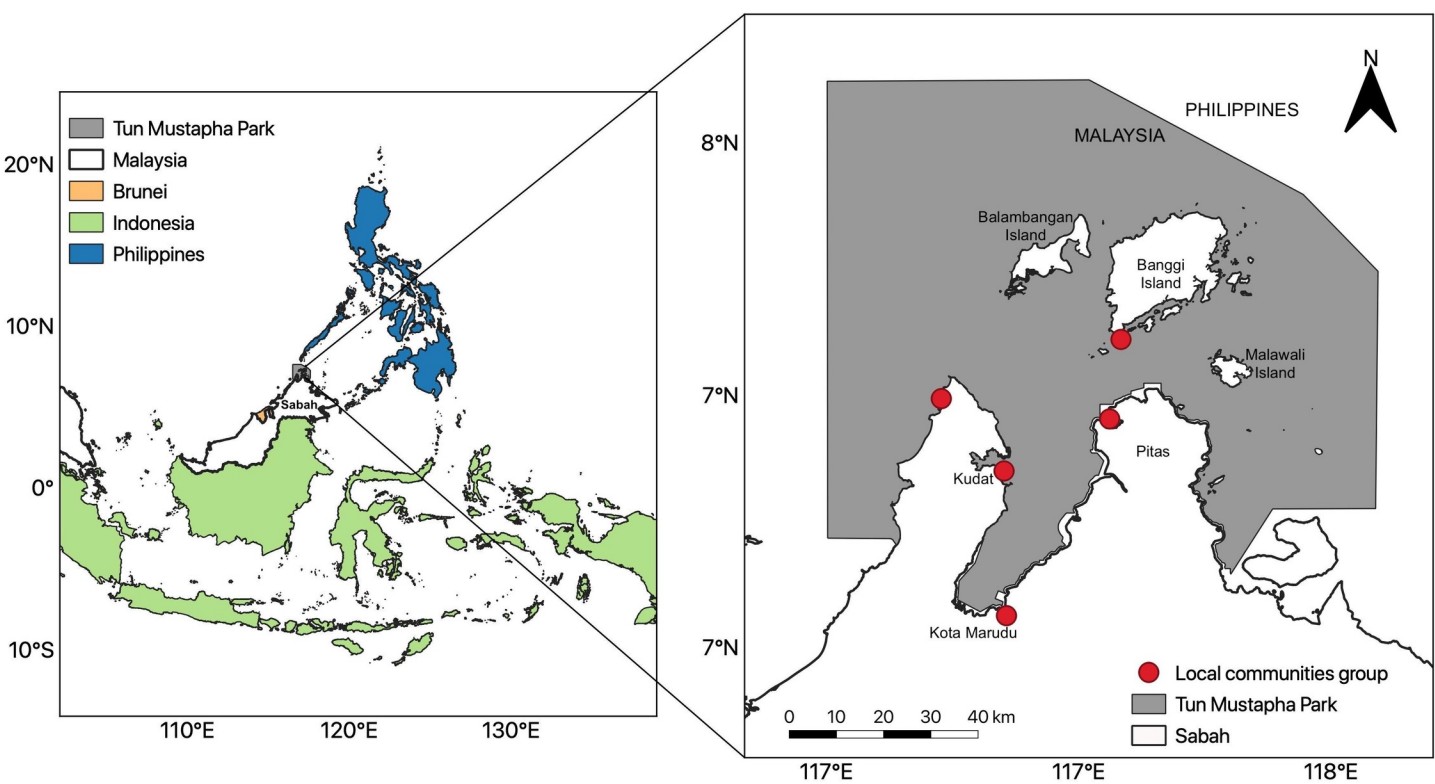

**Fig 1. The study site which is Tun Mustapha Park.** This study site is located along the international boundary between Malaysia and Philippines. Participants for this project were recruited from five community groups residing in the marine park. The spatial data was created using QGIS 3.3.3 based on the public dataset by Natural Earth (http://www.naturalearthdata.com/) and is for illustrative purposes only.

fall under the jurisdiction of Sabah Forestry Department under the Forest Enactment 1968, while other coastal areas including beaches are under the local District Offices.

## Ethics and permissions

The permission to conduct this work at TMP was obtained from Sabah Parks (Ref. No.: TTS/IP/100-6/2 Jld 11 (21)) and Sabah Biodiversity Council (Ref. No.: JKM/MBS.1000-2/2 JLD.8 (133)). The methodology for this work was approved by University of Malaya Research Ethics Committee (UM.TNC 2/UMREC-465). The design of this work is detailed in Fig 2. Informed consent from the participants were also obtained prior to the activities; full details are available in section "consent and considerations".

## Definition of ecosystem services and habitats of interest

To ensure participants could provide information that is appropriate to this work, it is important for them to have the same understanding of ES with us. Firstly, we conducted a reconnaissance survey [36] and a literature review (Lim et al., unpublished) to determine the categories of ES and habitats to be the focus for this work (Fig 2). Four categories of ES: (i) provisioning, (ii) cultural, and (iii) regulating and (iv) supporting services following definitions by TEEB [3] (Table 1) were introduced to participants during participatory mapping and pre-workshop for Photovoice, along with a handbook. Note that during the reconnaissance survey, our engagement with local communities suggested that they may use the terms "regulating service" and

- Literature review of habitats and ecosystem services in Tun Mustapha Park
- Reconnaissance survey (Boey et al. 2018)

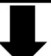

- Defined habitats and ecosystem services of interest
- Recruited participants via snowball sampling

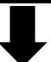

- Two rounds of participatory mapping with community-based organisations and mariculture farm

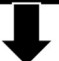

- Workshop to brief participants about the Photovoice activity, protocol, definition of ecosystem services and ethical considerations

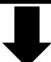

- Two rounds of Photovoice activity with community-based organisations

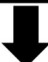

- Map digitisation and thematic analysis

**Fig 2. Workflow of this study in summary.**

"supporting service" interchangeably. Nevertheless, the information provided by the participants could be categorised into the four categories of ES. While we acknowledge the diverse marine-associated habitats in TMP, we asked the participants to consider the following habitats: beaches (i.e. sandy, muddy and rocky), corals, mangroves and seagrasses to guide them in mapping the ES. Following the definition and categories of ES by Millennium Ecosystem Assessment [2] and TEEB [3], we considered the information provided by the participants in

**Table 1. Categories of ecosystem services used here that are related to the marine ecosystem in TMP.** Categories, definitions and examples of the ecosystem services were introduced to participants before the start of activities to ensure they could provide information suitable for this study. The definitions and examples follow the glossary provided by TEEB [3].

| Category of ecosytem service | Definition | Example of benefits from ecosystem service |
|---|---|---|
| Provisioning | Material derived from nature, animals and plants for human use and habitats used for agricultural and aquaculture purposes | Food, beverage, medicine seeds for agriculture and aquaculture |
| Cultural | Non-material benefits derived from nature for spiritual and mental well-being in individuals as well as beliefs and knowledge within communities. The knowledge includes educational opportunities [2]. | Recreation, tourism, craft making, inspiration for culture including local myths, folklores, activities and skills pertaining local customs, traditions and religion, community-based spiritual experiences and belonging |
| Regulating | Natural processes that regulates and maintain the environment which benefits human | Climate regulation, sea erosion prevention, carbon storage, pollination, and treatment of toxic elements |
| Supporting | Natural processes that underpins almost all services (mentioned above), specifically in maintaing the habitats for supporting the population and diversity of flora and fauna | Genetic diversity of commercial organisms, and grounds for mating, nesting and feeding for charismatic and commercial organisms |

regards to human activities (e.g. recreation), goods (e.g. seafood) and services (e.g. nesting and feeding grounds for commercial species) in a specific habitat as benefits they received from ES.

## Participants

We employed snowball sampling to recruit participants, where our initial contacts introduced us to potential participants who further recruited other participants [37]. Our initial contacts were past and current members of government agency and non-governmental organisation (NGO) whom we contacted through official communication. Between August 2018 and February 2019, we were introduced to four community-based organisations (CBO) and one mariculture farm by our initial contacts (Fig 1). The CBO in Pitas and the mariculture farm in Kudat did not participate in the establishment of TMP while the others did. Members of these groups were residing and/or working in the marine habitats of our interest in TMP, and hence represent the "primary users" of the ES. We first met the leaders of the CBOs and the owner of the mariculture farm, where we introduced ourselves formally, proposed our research activities (i.e. participatory mapping and Photovoice) and explained the implications of their participation in supporting the existing management of TMP. Subsequently, the leaders and farm owner recruited their members and employees respectively to participate in both activities. Although we reminded the leaders and farm owner to balance the demographic profile of their invitees (e.g. ratio of gender and age), the participants of participatory mapping were mostly older and male whereas there were more younger and female participants in Photovoice. Socio-demographic details of the participants were available in Table 2.

Participatory mapping was conducted in 2019 with the CBOs and mariculture farm whereas Photovoice was conducted in 2020 with the CBOs only as the mariculture farm did not respond to our invitation. Note that only two of all participants who joined the Photovoice also joined the participatory mapping while the rest did not join the latter.

## Consent and considerations

Prior to the activities with participants, consents were firstly obtained from the leader of CBOs and owner of mariculture farm. All participants, who were either member of the CBOs or

**Table 2. Socio-demography of participants who joined the participatory activities.**

| Activity | Date | Cohort | Participant | Age range | Ethinicity | Employment |
|---|---|---|---|---|---|---|
| *Participatory mapping* | | | | | | |
| First round | April 2019 | 4 CBOs and 1 mariculture farm | 22 people (male = 20, female = 2) | 25 to 62 (mean = 39) | Bajau (27.27%), Rungus (18.18%), Suluk (18.18%), Sungai (13.64%), Bajau Suluk (4.55%), Benadan (4.55%), Chinese (4.55%), Kagayan (4.55%) and Filipina (4.55%) | "fishermen" (45.45%), "small trading business" (22.73%), "waiter" (9.09%), "odd job" (4.55%), "scuba diver" (4.55%) and "farmer" (4.55%). The remaining 9.09% did not state their employment. |
| Second round | June 2019 | 4 CBOs and 1 mariculture farm | 20 people (male = 17, female = 3) | 30 to 68 (mean = 47) | Bajau (30%), Sungai (25%), Rungus (15%), Suluk/Sungai (5%), Bajau Samah (5%), Bajau Ubian (5%), Dusun (5%) and Chinese (5%). The remaining 5% did not state their ethnicity. | "fishermen" (45%), "self-employed" (15%), "being employed" (5%) and "farmer" (5%). The remaining 30% did not state their employment. |
| *Photovoice* | | | | | | |
| First round | January 2020 | 3 CBOs | 8 people (male = 3, female = 5) | 20 to 45 (mean = 25) | Suluk (50%), Sungai (12.5%), Dusun (12.5%), Kadazan Dusun (12.5%) and Brunei-Melayu (12.5%) | "student" (25%), "teacher" (12.5%), "small trading business" (12.5%), "being employed" (12.5%) and "self-employed" (12.5%). The remaining 25% did not state their employment. |
| Second round | February 2020 | 4 CBOs | 8 people (male = 1, female = 7) | 20 to 45 (mean = 29) | Suluk (50%), Rungus (25%), Sungai (12.5%), and Brunei-Melayu (12.5%) | "self-employed" (50%), "receptionist" (25%) "teacher" (12.5%) and "housekeeper" (12.5%). Remaining 30% did not state their employment. |

workers at the farm, were briefed about the objectives of the activities, type of activities to be conducted with them, their voluntary and anonymous participation, and their rights to withdraw should they feel uncomfortable. A copy of an information sheet containing details of the activities and contact details for formal complaint was given to the participant (S1 and S2 Files). The participants then completed a consent form, which also comprised demographic questions, to indicate that (i) they understood the nature of activities that they will be participating in, and (ii) agreed to join the activities on voluntary basis (S3 and S4 Files). The sessions with the participants were not recorded audibly and visually due to requests from participants. The briefing and documents were in Malay language which is one of the main languages used in TMP to ensure that the participants were well-informed about the activities that they were participating in. Following Masterson, Mahajan, & Tengö [28], we reminded the participants to refrain from entering places and photographing activities that will harm themselves. We also informed the participants that any photographs of recognisable individuals doing illegal and destructive activities would not be used for publicity. If participants were impacted by this work (e.g. ostracised by other community members), the activities with them would be stopped immediately. We also reminded them that all information and photographs belong to participants, and they could decide whether their input to be used here.

## Participatory mapping

Two rounds of participatory mapping were conducted in the form of focus group discussion. Considering the power dynamics where some participants may hesitate to speak due to the presence of certain participants, we grouped the participants homogeneously (i.e. members of same CBOs) to encourage them to speak freely based on their similar experiences [38]. Each round was conducted by one moderator (who asked questions) and one facilitator (who assisted participants to answer the questions by drawing or writing). An A1-sized map of TMP was provided to each group of participants together with stickers of habitats and activities that may occur in TMP based on the literature review conducted prior to this step. To guide the

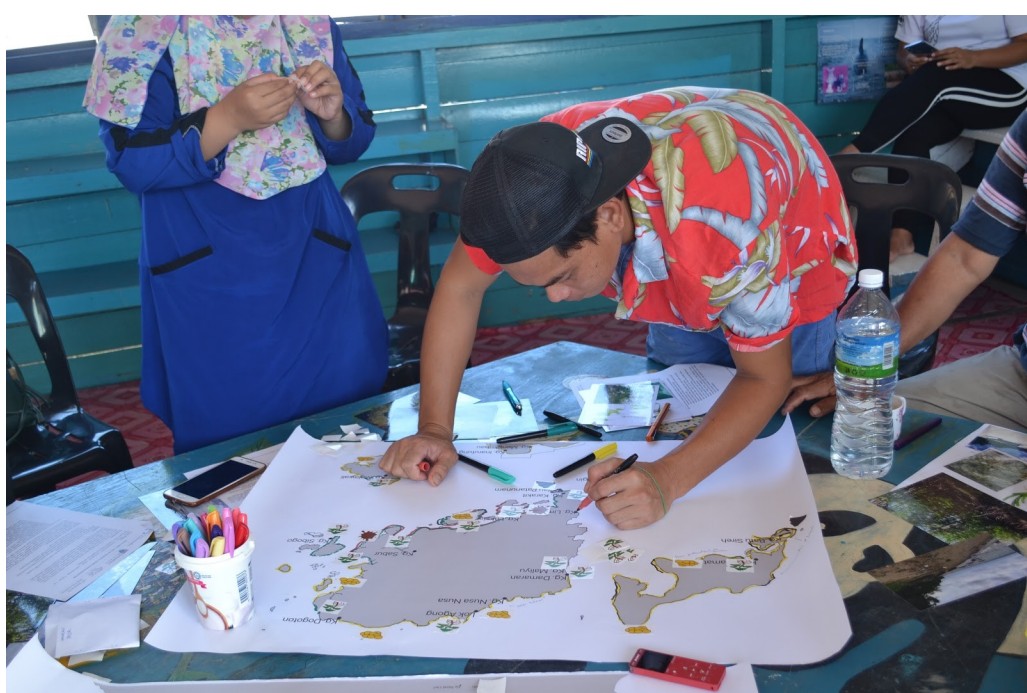

**Fig 3. Participants responded to the questions in participatory mapping.**

participants in mapping the services and habitats, name of villages were included in the TMP map and photographs of habitats (i.e. mangroves, seagrasses, corals, beaches) were shown to participants. The participatory mapping comprised three sequential parts, where participants were asked questions about (i) habitats that are associated with marine ecosystem in TMP, (ii) activities in the habitats that are beneficial to them, (iii) importance of the habitats that they mentioned and (iv) activities that are threatening the habitats (S1 Table). Participants responded to the questions by pasting stickers provided by us which represent habitats and activities [39] and drawing additional features using colourful pens on the map [24] (Fig 3).

After the first round of participatory mapping, the spatial input from the participants were digitised using an open source geographic information system application, QGIS 3.3.3 [40]. When digitising the maps, the input from participants were compared to satellite imagery in Google Earth Pro (www.google.com/earth) to cross-check the location of habitats. Locations of coral reefs provided by participants were overlapped with spatial data of corals in the region sourced from UNEP-WCMC, WorldFish Centre, WRI and TNC [41] to check the accuracy of the coral reefs' locations. As several activities and habitats overlapped and considering it is impossible to show all details in the map, we generalised some of the details (e.g. fishing and coral reefs distribution) on the basis of amalgamation, exaggeration and selection following Traun, Klug, & Burkhard [42]. During the second round of participatory mapping, the digitised maps were shared with participants to obtain their consensus for validating the information and subsequent analyses. The participants were asked if they agree with the maps and whether they have any feedback regarding the maps. We also asked the participants if they would like to elaborate further on the activities that they have located on the map. Further feedback from the participants were taken with no changes made to the maps.

## Photovoice

The participants first joined a 2-day workshop in December 2019 at a local terrestrial state park. This is to ensure that participants understand the protocol of Photovoice and conduct it in a safe environment besides minimising the bias effects where participants may photograph specific habitats and activities (e.g. fishing at sea) if the workshop were conducted within the marine park. The participants were introduced to the Photovoice activity, definitions of ES (Table 1), research ethics to be observed, basic photography, Google Maps on smartphone for GPS coordinates and examples from other Photovoice projects. The participants also attempted the Photovoice activity in the terrestrial park by using their smartphones to take photographs, write captions and obtain GPS coordinates.

After the workshop, participants were given fourteen days to photograph habitats of interest (i.e. mangroves, beaches, seagrasses, corals and sea in general) which are meaningful to them and their community following Bennett and Dearden [27]. The fourteen days period was to allow participants to have sufficient time to take photographs and articulate appropriate caption. For each photograph, participants must provide a caption, date and location. The following questions were provided to them as a guide:

1. What in the sea, mangroves and beaches that are important to you, your family and your community?

2. What are the changes to the sea, mangroves and beaches that have impacted you, your family and your community?

3. What do you, your family and your community hope for the future of the sea, mangroves and beaches?

4. What do you, your family and your community worry about the future of the sea, seagrasses, coral reef, mangrove and beach?

After fourteen days, we conducted a group discussion with the participants, where they shared their photos with us and filled in a form containing questions of mnemonic PHOTO and about their thoughts regarding their photos (S5 File and S2 Table). Mnemonic PHOTO comprised five reflective questions which were modified from Hergenrather et al. [43]:

1. What is the story behind your **P**hoto?

2. What are the ecosystem services **H**appening in your photo?

3. Why did you take a photo **O**f this?

4. What are the **T**hreats to your life or your community in this photo?

5. How can this picture provide **O**pportunities for things to be better in future?

We also encouraged discussion between the participants when answering the questions. The same process was conducted for the second round of Photovoice. Due to the Movement Control Order in Malaysia as a response to Covid-19 pandemic [44], all activities with the participants were ended in March 2020.

## Result

### Participatory mapping

When the digitised maps from the first participatory mapping were shown to the participants during the second round, all of the participants agreed with the details on maps and provided

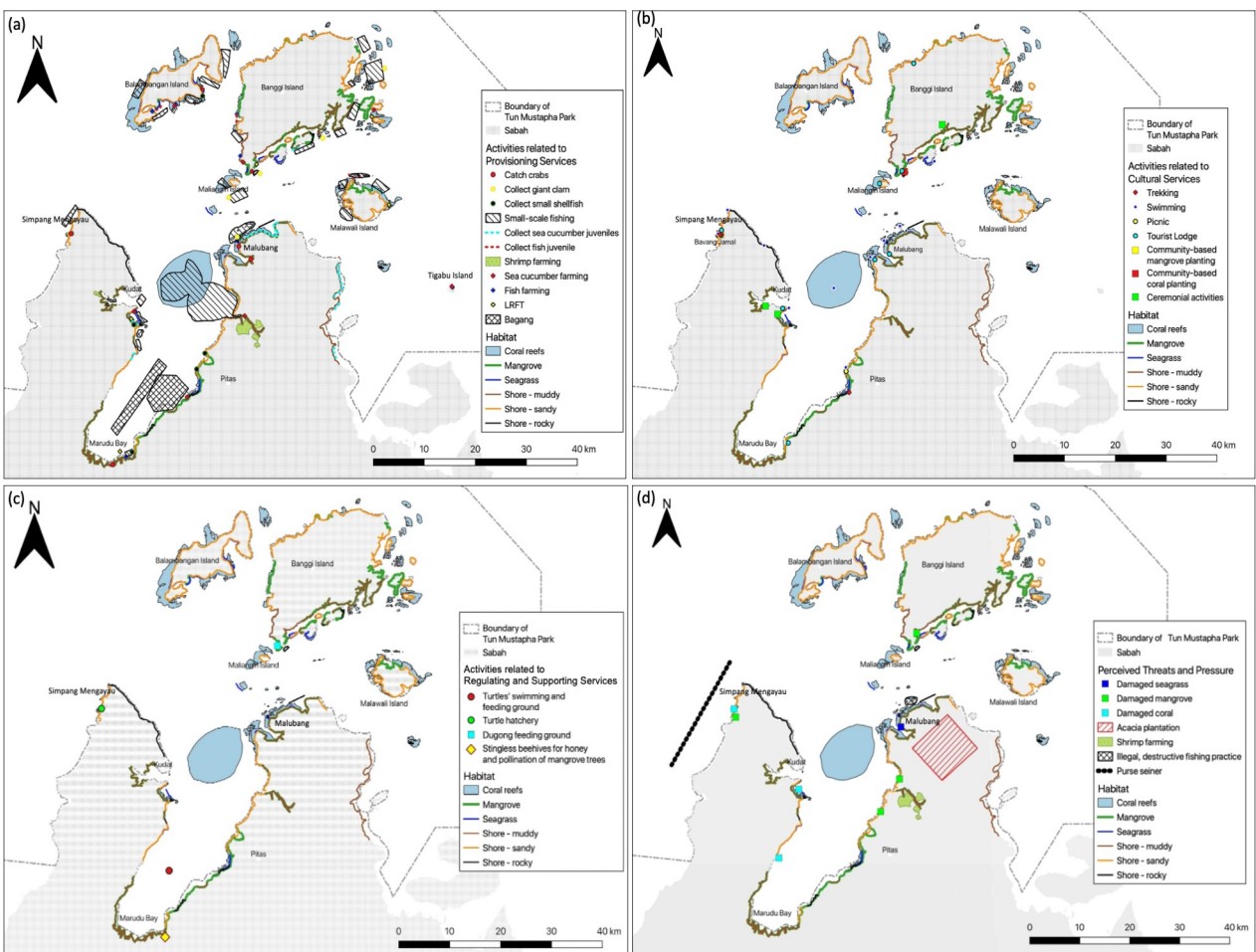

**Fig 4. Digitised maps resulted from participatory mapping showing marine-associated habitats, ecosystem services and perceived threats in Tun Mustapha Park.** (a) provisioning services. (b) cultural services. (c) regulating and supporting services. (d) perceived threats and pressures to the habitats and services. The information for each category of ecosystem service provided by the participants is considered as the benefits obtained from the marine ecosystem in TMP. The map was magnified to show the details. Note that there is a small island at the south of Maliangin Island called Maliangin Kecil Island which is surrounded by corals and seagrasses according to participants. LRFT stands for live reef fish trade. The map was created using QGIS 3.3.3, of which the distribution of coral reefs was referred to the public dataset by Ocean Data Viewer (https://data.unep-wcmc.org/datasets/1) and is for illustrative purposes only.

only elaboration of the activities that they have located on the maps such as the details of a folk-lore associated with the rock islands. No changes were made to the maps after the second round of participatory mapping. The participants also asked for a copy of the maps to assist them in understanding the marine-associated habitats near their village and identifying potential sites for ecotourism. Based on the maps and discussions from participatory mapping, we obtained the following spatial information for TMP: marine-associated habitats, provisioning services, cultural services, regulating services, supporting services, and perceived threats and pressures to habitats and services (Fig 4).

**Marine habitats in TMP.** "Beaches" were re-labelled as shores to include intertidal zones which are areas that are exposed to air during low tide and submerged during high tide (Fig 4). We observed that participants could distinguish and map the habitats that were considered in this study (i.e. beaches/shore, corals, mangroves and seagrasses). They also drew spatially extensive habitats (i.e. mangroves, muddy and sandy shores) on the map. When asked, the

participants responded that they knew about the habitats from their provisional jobs in catching seafood (i.e. fishing, gleaning), guiding foreign tourists, scuba diving and delivery service as well as from their leisure activities including visiting, swimming and picnicking. They added that mangroves were often associated with mud and therefore, coastal areas that were located near to mangroves were perceived by them to be muddy.

**Provisioning services.** According to the participants, seafood collection especially fishing were the main provisioning service in TMP, often occurring at where the corals are (Fig 4A). Small-scale fishing activities often took place near islands (i.e. Banggi, Maliangin, and Malawali) as fishers do not have to travel far from their village at coastal area. Collection of sea cucumber juveniles and crabs appeared to take place at muddy and sandy shores that appeared to be located close to mangroves. Giant clams (*Tridacna gigas*) were often collected at coral reefs whereas small shellfishes (without specific epithet) were often collected near sandy shore. "Bagang", a traditional static lift net which targets anchovies [45, 46], could be found in waters from Marudu Bay to Pitas. The participants added that the location was suitable for bagang as the mainland (i.e. Kudat and Pitas) provided protection to the bagang from strong wind, hence preventing the bagang from collapsing.

Mariculture farming of fish and sea cucumber took place near sandy shores, mangroves, coral reefs and seagrasses. The participants added that environmental factors (i.e. water quality and sedimentation) influence the farming. Mariculture farming also took place at Tigabu Island which they perceived to be safe due to the presence of a military base camp there. While shrimp farming was not exactly located in "marine habitats" but rather at riverine mangrove in Pitas (see [34]), the participants included shrimp farming as provisioning service here. They claimed that the water, which flows from riverine mangrove to sea, could influence the quality of the environment for mariculture farming.

**Cultural services.** The ecosystem in TMP provided opportunities for recreation, tourism and education. Participants highlighted three ceremonial activities where local fishermen worship natural rock outcrop near the shore for blessing before fishing (Fig 4B). Many recreational activities in TMP were participated by local communities as well as domestic and international tourists. Locals often picnic at sandy shores as well as swim and free-dive at coral reefs. Domestic and international tourists often visit the Supirak Island which is associated with a local folklore and the Floating Coral Bar that were made of dead corals for swimming, fishing and picnicking. The tourists usually take the boats operated by the locals communities at Malubang to visit these sites and stay at a homestay operated by the locals. Likewise, there was a private resort at Maliangin Island for international tourists. At Bavang Jamal, northwest of Kudat, locals and tourists usually swim at the coral reefs and trek at a riverine mangrove that connects to the estuary. There were also villas and homestays there for tourists who wish to spend the weekend at the sandy shore and for corporate companies to conduct team-building activities. The participants added that the communities in Bavang Jamal have initiated the planting of corals and mangroves in the area previously as part of community-based educational programme. Likewise, a local youth club, Banggi Coral Conservation Society (formerly known as Banggi Youth Club), had initiated coral planting activities as part of educational programme at the south of Banggi Island, but had stopped due to the lack of funding.

**Regulating services.** When asked, participants considered animals only despite the questions were framed to guide them to consider abiotic components too. They mentioned that there were beehives of stingless bees near Marudu Bay (Fig 4C), mostly as apiculture by locals, which play a role in pollinating the mangrove trees near the bay besides providing honey.

**Supporting services.** The participants located hatchery and foraging grounds for turtles on the map (Fig 4C). The hatchery was set up by a local youth club, Kudat Turtle Conservation Society (KTCS), whereas the locations of feeding ground were based on their observation

during fishing. According to the participants, dugongs (*Dugong dugon*) have been sighted feeding at the south of Banggi Island. They added that these areas are important for supporting the population of these animals.

**Perceived threats and pressures to habitats.**   Participants located several areas where they claimed the mangroves, seagrasses and corals have been damaged by human activities, though they did not elaborate on this (Fig 4D). They highlighted the tree plantation of genus *Acacia* near Malubang could endanger the fishermen nearby as the logs were transported to mill via floating transfer platform in the sea. Although the shrimp farming occurred at riverine mangrove at Pitas, the participants thought that it was responsible for damaging the mangroves and causing murky waters in that area. Participants also highlighted destructive fishing activities (i.e. compressor fishing method and hammering the giant clams) occuring at northwest of Malubang which often involved outsiders (i.e. not members of local communities) encroaching their fishing ground. Purse seiners along the northwest of Kudat were perceived by the participants as threat due to the method which catches fish indiscriminately.

## Photovoice

Sixteen photographs were provided by the participants of the Photovoice activity. Based on the captions and discussion among the participants during our meeting with them, we were able to elicit further details of ES and perceived threats (Table 3). When analysing the captions and their discussion points, we identified three common themes namely "Environment" (seven photographs; Fig 5), "Sociocultural" (six photographs; Fig 6), and "Economics" (three photographs; Fig 7), which encapsulated the ES categories of our interest.

**Environment.**   Participants depicted issues of degrading environment that were associated with cultural, provisioning, regulating and supporting services in their photographs together with captions (Fig 5). One participant wrote that sharp mangrove stumps were often washed ashore during wet season near Marudu Bay, which could endanger beachgoers and tear fishers' nets (Fig 5A). However, these mangrove stumps have values of cultural services as they can be used for woodcarving and landscaping. Other participants highlighted the adverse impacts of anthropogenic activities on mangroves, shores and seafood supply, which in turn affect the provisioning services and well-being of the local communities (Fig 5B, 5F & 5G). The participants also mentioned the role of mangroves and a river connecting to the sea in providing regulating and supporting services, which were soil stability and habitat for other organisms (Fig 5C, 5D & 5E). When asked for their suggestions to improve the situation depicted by their photographs, the participants responded "educational awareness programmes", "community-based cleaning activities", "systematic waste disposal", "ban of destructive activities", "establishment of protected areas", and "enforcement of fishing regulations".

**Sociocultural.**   The sociocultural aspects of participants' life depicted in the photographs were associated with cultural services. Sandy shores, coral bars, rock islands provided opportunities for local communities as well as domestic and international tourists to engage in recreational activities such as appreciating sunset and rock formations (Fig 6). The rock islands near Malubang, Pitas were associated with a local folklore, where a man named Supirak was cursed by his mother to turn into a stone because he disowned her after becoming successful with fame and wealth. A rock island in Malubang named Supirak Island, which shaped like a ship, was used by Supirak for sailing according to the local belief (Fig 6D). Another rock island which is also a tourist destination near Malubang is Batu Berunsai which shares the name with a traditional dance of local ethnic groups Suluk, Bajau and Kagayan (Fig 6E). One participant wished for an isolated village near Marudu Bay, which depends on marine resources for livelihood, to be shared with knowledge of sustainable use of natural resources so that the villagers

**Table 3. Ecosystem services depicted by the photographs, accompanying captions and responses to questions by researchers during group discussion as resulted from the Photovoice activity.** Note that the information for each ecosystem services provided by the participants should be considered as the benefits they received from the ecosystem services.

| Benefits from ecosystem services | Associated habitat | Perceived threat |
|---|---|---|
| *Provisioning* | | |
| Source of protein (i.e. fish, crab, prawn, shellfish, cuttlefish) for subsistence and sale | Shore | Litter |
| | Mangrove | Clearing of mangroves |
| | Coral reefs | Riverine mangroves drying up during dry season |
| | Sea in general | Illegal and unregulated fishing activities |
| | | Fish bombing |
| Fuel | Mangrove | Unregulated cutting and pollution |
| Material for arts and crafts (i.e. wood stump) | Shore | |
| Material for construction (i.e. pillar) | Mangrove | |
| *Cultural* | | |
| Recreation and ecotourism | Coral bar | Fish bombing |
| | Shore | Erosion |
| | Rock islands | Litter |
| | Sea in general | |
| Creative activities (i.e. photography) | Shore | Litter |
| Education and research | Mangrove | Clearing of mangroves |
| Traditional knowledge, myth and belief | Rock islands | Litter |
| Landscape appreciation (i.e. sunset and unique shape of wood stumps) | Shore | Fish bombing |
| *Regulating* | | |
| Prevention of sea erosion | Coral bar | Fish bombing |
| | Mangrove | |
| Regulation of air quality and surrounding temperature | Mangrove | Unregulated cutting and pollution |
| *Supporting* | | |
| Shelter and nesting ground for wildlife | Rock island | Erosion |
| | Mangrove | Riverine mangroves drying up during dry season |
| | Sea in general | Illegal and unregulated fishing |

could fish sustainably and "systematically" (sic) in order to maintain the long-term supply of natural resources which is also the source of their livelihood (Fig 6F). When asked for their suggestions to improve the situation depicted by their photographs, the participants responded "enforcement of regulations pertaining to fishing, waste disposal, encroachment and patrolling" and "provision of public facilities for recreational activities including dustbins and huts".

**Economics.** The participants associated the economic aspect of their life with provisioning and cultural services (Fig 7). In Malubang, the communities depend on marine resources for subsistence and livelihood (Fig 7B & 7C). According to the participants, the communities used to fish for a living but mostly were shifting to tourism related jobs due to the popularity of the Supirak folklore and recreational opportunities in the area. The communities were offering recreational activities (e.g. kayaking, swimming and angling), guides (e.g. local folklores associated with rock formations), scenic views (e.g. sunset and unique formation of rocks) and homestay experience (e.g. traditional fishing methods and local cuisines) to domestic and international tourists (Fig 7A & 7B). Therefore, the communities in Malubang were motivated

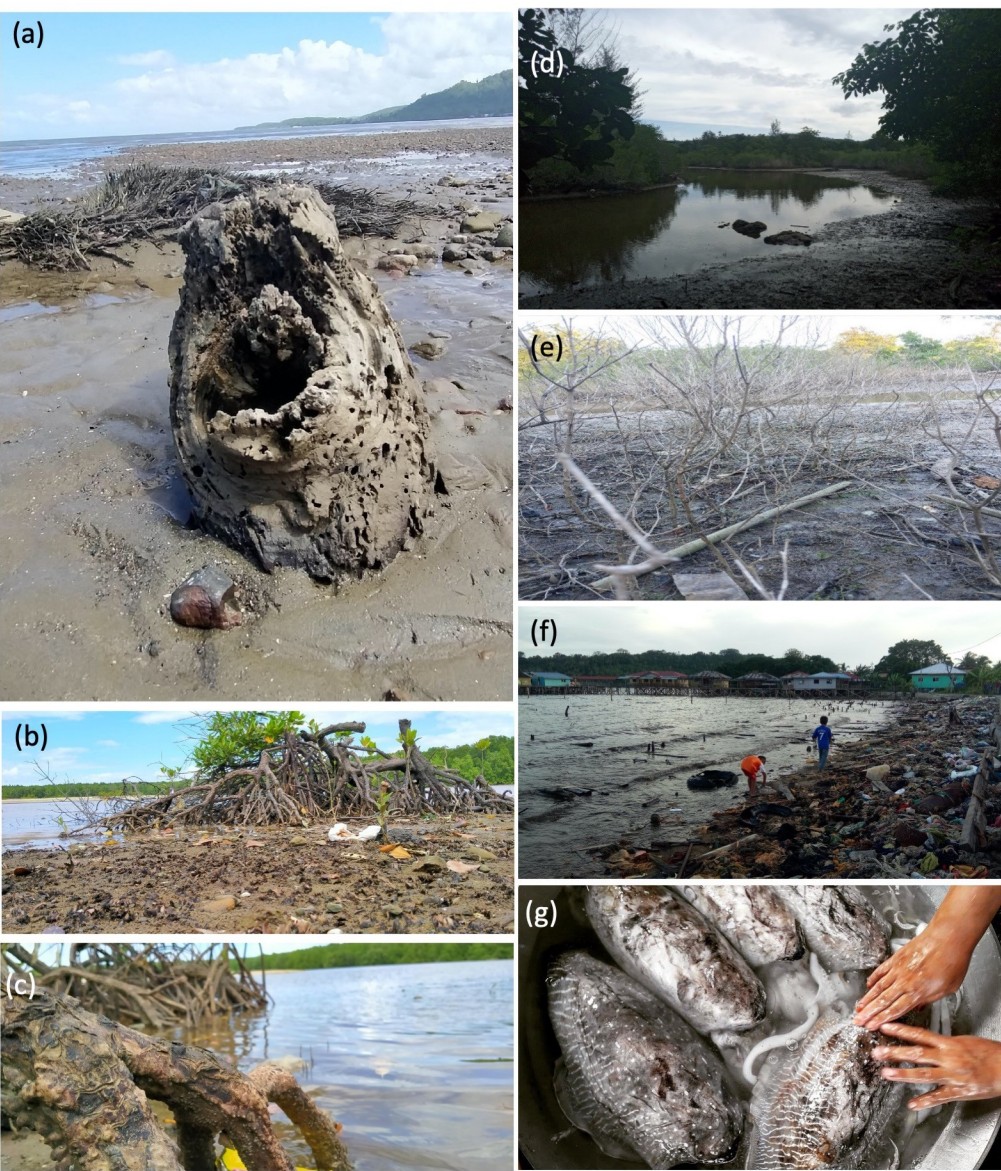

**Fig 5. Photographs portraying environmental aspects of participants' life associated with cultural, provisioning, regulating and supporting services.** (a) Mangrove stumps were washed ashore during wet season and can be used for craft. (b) Unregulated cutting and pollution could cause the extinction of mangrove trees. (c) Unique looking roots of mangrove trees could strengthen the soil structure and prevent sea erosion at shore. (d) This river at a mangrove was experiencing erosion and becoming shallower, hence causing population decline of its inhabitants. (e) The river at this mangrove was drying up and have poor water flow, causing the trees unable to grow and hence died. (f) The poor waste disposal system had caused the accumulation of rubbish at this shore, causing discomfort to the villagers. (g) Cuttlefish is a local favourite seafood which used to be abundant in coral reefs surrounding the Banggi Island but is rarely caught nowadays due to uncontrolled fishing activities.

to conserve and protect the habitats from destructive activities due to the economic values of the ES. When asked for their suggestions to improve the situation depicted by their photographs, the participants responded "clearing rubbish from habitats", "enforcement of regulations pertaining to littering, destructive fishing methods and littering", and 'educational awarenes programmes for destructive fishing methods".

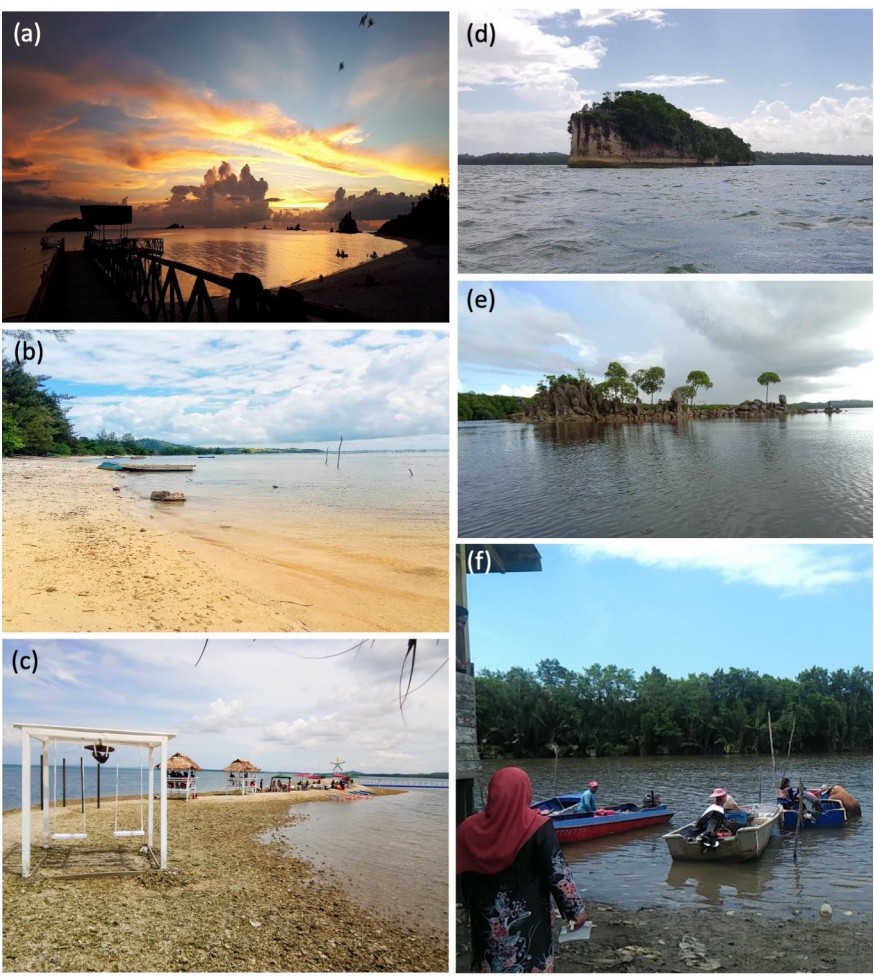

**Fig 6. Photographs portraying sociocultural aspects of participants' life associated with cultural services.** (a) This scenic view of sunset makes the shore at Banggi Island an attraction for domestic and international tourists. (b) This shore at Banggi Island is where fishers dock their boats and families do recreational activities. (c) This Floating Coral Bar near Malubang is a tourist destination in TMP. (d) Supirak Island is a rock island that has a local folklore and potential for tourism. (e) Batu Berunsai, which is another rock island near Malubang, is a tourist destination and is related to a local dance. (f) An isolated community residing in mangroves near Marudu Bay that should be shared with knowledge of sustainable natural resource use.

## Discussion

An effective management of natural resources and protected area requires the participation of local communities and the incorporation of their local ecological knowledge, as such knowledge represents a close relationship between the communities and their natural surroundings [13]. Such relationship is significantly important when local communities utilise the goods and services provided by the ecosystem for subsistence and livelihood [15, 28]. Therefore, local communities may possess ecological knowledge that is exclusive to their community yet crucial for managing natural resources and protected areas [17]. Here we reported the findings of community-partnered participatory research in a newly established multiple-use marine park, which could further support decision-making and existing management plan for the park.

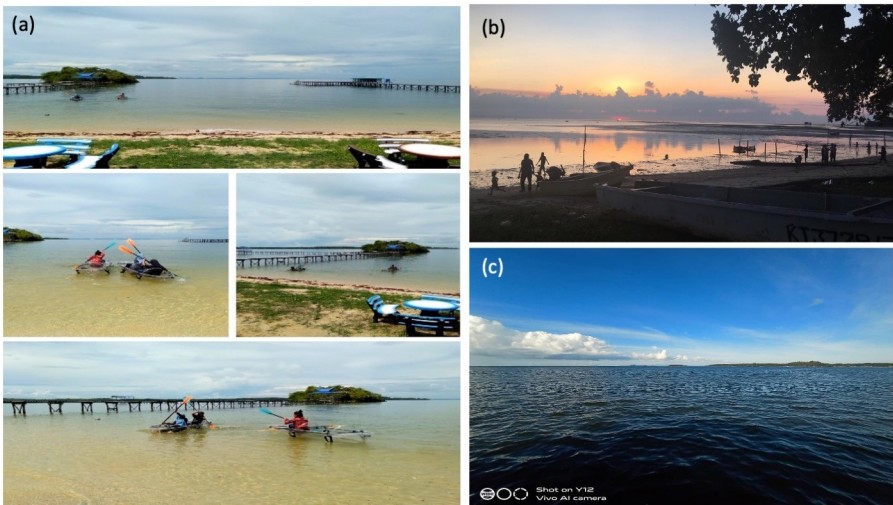

**Fig 7. Photographs portraying economic aspects of participants' life at Malubang village associated with cultural and provisioning services.** (a) This collage shows a shore that was a destination for tourists to do recreational activities, hence providing business opportunities for villagers. (b) This shore was a destination for tourists and villagers to do recreational activities such as swimming and photography besides being source of seafood. (c) This sea near Malubang was a source of livelihood for villagers as they fish here.

## Values of ecosystem services

The communities in TMP mostly valued the marine ecosystem for economic purposes. Findings from participatory mapping mostly focused on provisioning services, particularly the seafood supply. This was predicted as the communities in the marine park mostly depend on marine resources for subsistence and livelihood [32]. The participants also shared that the damage to mangroves and corals resulted in fewer mature fishes available to catch, as well as water-transported logs resulted from acacia planting were endangering fishers in the sea. On the other hand, Photovoice revealed that the communities were slowly shifting from fishing to tourism and recreational-related jobs due to the opportunities provided by the ES. Recreational activities and appreciation of landscapes were categorised under cultural services [2, 3], but these were being utilised by communities as benefits from provisioning services. For a small fee, the communities offer recreational-related services (e.g. guides, kayaking, homestay experience and boat services) to domestic and international tourists.

Cities nearby often contribute to large demand of provisioning and cultural services in a protected area [6]. In addition, presence of specific landscape and species are crucial for determining the value of the services, especially the cultural services [47]. Here, participants shared that the tourists residing outside TMP are the major demand of the cultural services, especially the appreciation of landscape in the marine park. Many homestays and tourist lodges were located at the habitats that provide scenic views, such as sandy shores and unique rock formations at Malubang and Simpang Mengayau, where guests can swim, snorkel and free-dive. The guests of these lodges were usually not resident of TMP. Furthermore, a local youth club (KTCS) offered public watching of turtle hatchling release at the sandy shore of Simpang Mengayau for a conservation fee, which had attracted those residing outside TMP.

When asked about their perception of regulating and supporting services during participatory mapping, participants tend to focus their responses on animals instead of abiotic features. This may be due to participants being able to relate better to animals that form part of their folklores and belief, such as the dugongs in Kudat and Banggi Island [48]. Note that the

participants of participatory mapping were mostly senior members of the community, and it is possible that regulating and supporting services were not central to their qualitative narratives of their environment and communities due to them prioritising provisioning services instead [27]. Conversely, participants of Photovoice were able to provide more details of regulating and supporting services in their photographs and captions. These participants are in their youth with many just completed their secondary education, besides having attended the pre-workshop of Photovoice where they were introduced to the ES. Education could play a role in promoting better knowledge of ecology and hence ecosystem services [49]. Therefore, participants of Photovoice who have received ecological information from researchers and possibly schools could provide more details of the regulating and supporting services that are less straightforward compared to the provisioning and cultural services.

External demand for ES may inflict negative impacts on the ecosystems and services in the area [6]. Locals have been reported selling unique-looking corals and shells as souvenir and curio to tourists at Simpang Mengayau, hence unregulated collection may have adverse impact on the corals and molluscs [50]. Furthermore, shores that were promoted as tourist destination may be more vulnerable to plastic debris pollution due to litter and waste generated by tourists [51]. In Banggi Island, the shores were heavily contaminated by plastic waste due to lack of proper waste disposal system as revealed by Photovoice. Promoting this area as tourist destination may contribute to generation of more plastic debris which will further damage the habitats and hence the well-being of the communities. Moreover, the participants perceived the shrimp farming which caters for external seafood demand at riverine mangrove located outside the border of TMP as a threat that affects the quality of the riverine water that eventually flows into the TMP's sea. Such border effect is an example of how anthropogenic activities surrounding TMP could threaten the marine park's capacity of supplying ES. As the ES in TMP carry significant social and economic values, identifying the threats and pressures within and outside the marine park to the ES could facilitate the mitigation of adverse impacts on the communities as well as framing the appropriate measures to protect the ES [6].

## Participatory mapping and Photovoice

Participatory mapping coupled with group discussion integrates the perceptions, knowledge and values of ES onto maps to assess the spatial distribution of the services within an area [12, 52]. Using maps allows the participants to highlight trade-offs and conflicts between the services and other land uses, often with a consensus view [29]. Conversely, Photovoice allows people to express their thoughts, experience and certain aspects of their life associated with their natural surroundings in photographs with in-depth qualitative captions that could be shared beyond their communities [26, 27, 53]. Photographs are self-explanatory and allow participants to highlight issues anonymously as it is an individual activity compared to group-based activity where participants have to voice openly [26].

Here, there were higher number of participants, mostly male, in participatory mapping compared to fewer participants, mostly female, in Photovoice. Although we have attempted to balance the ratio of gender among the participants by reminding the leaders of CBOs and farm owner, such gender differences in the two activities may indicate cultural and social barriers. As seen in Kenya and South Africa, there are cultural expecations of women being modest and hence, women have little opportunities to voice out and involve in any decision-making within the community [1, 28]. Such cultural and social barriers may have impeded the participation of local women in our participatory mapping which is a group-based activitiy, as they may not feel comfortable in voicing their thoughts in public. Conversely, Photovoice is an individual activity which required the participants to fill in a questionnaire themselves with their

photographs as reference and this may have allowed them to pen down their individual thoughts. Therefore, Photovoice may have provided the women the confidence to document their views in photographs and narratives [28]. We also found that having a group discussion after the Photovoice activity allows the female participants to exchange views and provide support to their thoughts of the photographs and captions.

We observed that male participants tend to provide information related to economic activities, particularly fishing and mariculture farming, during the participatory mapping. In comparison, female participants focused on environmental issues during the Photovoice activity such as plastic waste at beach as well as education and photography opportunities. This suggested that men and women have distinctive interests and needs for the ES; in this case, income versus nature appreciation [54]. Furthermore, men and women are likely to interact with different parts of the marine ecosystem due to their roles, where men fish in the sea while women perform their household duties at the coastal area [55]. Women also tend to value a wider range of ES, particularly those that socially benefit the households and communities [55]. Such gendered differences in the perception of ES indicated the importance of participation of women in decision-making when managing marine protected areas.

During participatory mapping, we observed few individuals dominating the group-based activity and therefore, we may have not captured all participants' actual thoughts. Furthermore, we noticed that participants tend to provide information of a locality, which was attributed to the (i) limited knowledge by younger participants (in their twenties) who have only visited certain locations in the marine park and (ii) domination of discussion by senior participants (in their fourties to sixties) who were interested in certain locations near their village. Although we have attempted to homogenise the demographic profile of the participants for the activities (i.e. inviting members of specific age and gender), male senior members of CBOs oftentimes participated in our activities usually out of curiosity and invited by the junior members.

One factor that encourages the participation in citizen science projects is the existence of user-friendly technical tools for data collection [22, 23]. Here, Photovoice requires participants to photograph certain activities or issues with their own smartphone and write descriptive captions within fourteen days before meeting the researchers. When asked, many participants admitted that they did not take photographs immediately after meeting the researchers as they need to look for inspirational places and events to help them with the Photovoice activity. For example, a group in Kota Marudu travelled to a particular coastal mangrove during the weekend for the Photovoice activity. Furthermore, photographing ES may be difficult and could not fully fit the stories or issues to be highlighted by participants [26], though we have included group discussions and questionnaire to facilitate the participants in expressing their thoughts. Photovoice also requires continuous engagement from researchers and commitment from participants. Although we tried to engage with participants frequently, there were two male participants from the participatory mapping who expressed their interest in the Photovoice activity failed to provide any photographs nor captions with the reason given being "busy". As fewer participants agreed to meet-up during the second meeting for Photovoice coupled with Movement Control Order in Malaysia due to Covid-19 pandemic [44], we ended the Photovoice activity after two rounds.

In contrast, participatory mapping only required participants to provide their thoughts immediately during the meeting with researchers, who also guided them in pasting the stickers and drawing the features on maps. This required lesser effort from the participants, which may explain the higher turnout of participants and more information shared by the participants for the participatory mapping. As demonstrated here, the success of participatory projects depends largely on the amount of participants' enthusiasm and whether the approach

employed is user-friendly. Therefore, we recommend participatory mapping which could attract more participants and require less effort from them. Nevertheless, Photovoice could supplement the information from participatory mapping as demonstrated here. As Photovoice activity requires commitment from participants, such approach would be suitable for promoting participation of minority groups (i.e. female in this case) in any stakeholder engagement and decision-making. However, more time would be required for Photovoice to generate the desired information as participants need to search for inspirational places and events.

As discussed above, participatory mapping allows inclusion of more people to capture more collective thoughts whereas Photovoice provides in-depth individual knowledge that supplements the consensus resulted from participatory mapping [1]. The two activities generated many similar information of activities that benefit from the ES, particularly fishing and recreational activities. However, there were information which were generated from the Photovoice that were not captured by the participatory mapping, such as the regulating services. Furthermore, gendered differences were observed which can be attributed to the domination of the participatory mapping by male participants whose feedbacks focused on economic activities, while the Photovoice activity was dominated by female participants who tend to voice their environmental concerns. It is unclear whether such differences were due to the two approaches employed here or influenced by the socio-demographic factors. The participants of Photovoice were mostly young female and have received ecological knowledge from the researchers and possibly from school. Note that the participatory mapping was also dominated by older male participants who were considered as senior members of the communities. Examining the factors that lead to the differences would require further employment of the two approaches in other communities with larger sample size that has a balanced socio-demographic profile. Nonetheless, when conducting group-based activities such as participatory mapping and group discussion, participants should be segregated based on gender, age and education as well as position in a particular organisation or community to minimise the domination of discussions and hence bias in feedback as well as to capture the different social dimension of ES [12, 38, 54].

The combination of several participatory approaches could enable the conversations among participants which carry political and community empowerment effects [12]. Here, the use of participatory mapping and Photovoice had allowed participants to highlight issues of encroachment into fishing grounds, destructive fishing practices and waste disposal system. This also allowed us to understand the conflicts of resource use, when participants claimed that the fishing ground at the northern Pitas had been encroached by individuals that do not belong to their community. The participants also provided suggestions for the issues such as awareness programmes, regulations for fishing and waste disposal, and strategies to boost local tourism. Moreover, they discussed the potential of additional livelihood (i.e. tourism-related businesses) as an alternative to fishing to improve their quality of life while conserving the marine resources.

Such information are useful to policymakers and authorities involved in decision-making pertaining to the conservation of particular areas within the marine park, but requires all stakeholders including communities and authorities to communicate and collaborate [6, 12]. For example, few participants attributed the poor plaste waste disposal system and irregular waste collection in Banggi Island to the absence of a representative of the local municipal in the island and lack of communication between the local municipal with the local communities. Yet stakeholder engagement which involved local communities, local government, waste operators, industry players and academic experts is pivotal for developing strategic waste management with examples from the UK [56] and Thailand [57]. We recommend the local municipal to actively engage with the communities and right agencies through interview, questionnaire

and focus group discussion to identify the issues, promote public responsibility and develop proper waste disposal system to prevent the plastic waste from further damaging the shores and affecting the well-being of the communities.

## Ability to provide information

Information from the participants matched with the information from our literature review, supporting the consideration of incorporating local knowledge in the management of protected area. For example, the participants highlighted the feeding ground of dugongs at southern Banggi Island (Fig 4C), congruent with Rajamani and Marsh [58]'s sightings of the animal within the area. The participants also highlighted the issue of waste disposal in Banggi Island (Fig 5F) which was also reported by Teh and Cabanban [59]. However, there were instances where participants provided inaccurate information such as "coral reefs" between Kudat and Malubang (Fig 4) which were not reported by Jumin et al. [35]. Some prior information did not surface during the participatory activities such as tourist lodges in Kudat that were observed during our reconnaissance survey [36]. Here, participants had provided reliable GPS coordinates for their photographs using smartphones. When the GPS coordinates were entered in Google Earth, we observed that the habitat in the satellite imagery matched with the habitat in the photographs and maps from this work, hence supporting the capability of participants in data validation. Most of the GPS coordinates were near villages, suggesting that participants collected data at their convenience. It is likely that the participants provided details of habitats that were near to them or frequented by them due to their familiarity of the area. Consequently, some information were aggregated within a small area and we could not retrieve larger spatial information. Therefore, we advise to incorporate local knowledge in the spatial planning and management of protected areas wisely due to potential human errors and biases.

Recruiting the right participants who are familiar with the locality of our interest is crucial for obtaining the maximum amount of information from the participants. This was possible through snowball sampling where our initial contacts introduced us to the participants who further recruited other participants [37]. Such approach is helpful when engaging rural communities that tend to be wary of outsiders [54]. Our choice of snowball sampling here had helped to placate some of the senior members during our first engagement with the CBOs. Furthermore, this approach had allowed us to recruit undocumented citizens in Banggi Island and illegal foreign workers in Kudat, who have provided information that were not given by other local participants (i.e. location of sea cucumber juveniles and certain coral reefs).

We initially planned for three group discussions for each group of participants for each activity, but we noticed that the amount of information provided by participants had reached saturation by the second round of the group discussion. As discussed by Priess & Kopperoinen [19], two to three rounds of focus group discussion could provide detailed information which require investment of time and costs to meet the participants, though one focus group discussion may be sufficient for a quick qualitative mapping as seen in this case. Other possible explanations to the saturation of information are having the same participants in each group discussion and the participants being fatigue with the repeating same exercise.

The choice of ES categories and definition along with habitats used in this work may have failed to capture some aspects of how communities benefit from ES within the marine park. One example is tourism that fits into our definition of cultural services but was being utilised by the communities as opportunities for income and thus could be considered as provisioning services. Moreover, based on our understanding, the present conceptualisations of ecosystem services [2, 3] do not consider the indirect impacts of perceived threats and conflicts within a protected area. Oftentimes, the establishment of protected area is to conserve the habitats that

provide crucial ES that benefits the communities within, and such purpose plays an important role in attracting financial support from external donors [1]. In our case, a youth club in Banggi Island acknowledged the impact of fish bombing on the coral reefs (i.e. habitat) and consequently on the supply of seafood and well-being of the fishers (i.e. ES). Therefore, the youth club approached external donors to support their initiative of replanting corals surrounding the island to ensure the long-term supply of seafood, though this initiative had stopped due to the lack of funding. Further understanding of the cascading effects of ES in a multifaceted context is required to determine the direct and indirect benefits provided by the ES and how this could improve our conceptualisations of ES.

## Implications for management

The main driver of the gazettement of TMP is the unsustainable fishing activities, with the aim of minimising the impacts of fishing activities through establishment of four zones to cater different scales of activities [31, 32, 35]. To reduce the local communities' reliance on fishing activities, the state government has been actively promoting ecotourism as an additional livelihood for the communities through homestay, tour and recreational activities [60]. This effort appeared to be well received by the communities as our participants provided information of the recreational activities and sites during both participatory mapping and Photovoice. Furthermore, our participants also requested for the digitised maps to assist them in planning and identifying potential sites and activities for ecotourism. With the maps provided to them coupled with their own photographs from the Photovoice activity, the participants could convince other members of their community to promote sustainable community-based tourism while preserving the integrity of the marine ecosystem. One example from Donsol Island, the Philippines demonstrated how the local communities protect whale sharks as their tourism asset which attracts tourists and hence boosting the local economy through food and accommodation businesses [61]. Nevertheless, this effort will require support from the local government such as providing training (e.g. sustainable ecotourism), funding and other forms of help (e.g. permits and infrastructure) to ensure the success of sustainable community-based tourism.

During the process of the TMP's gazettement, the CBOs were involved to provide their feedbacks except for the CBO that is based in Pitas and the mariculture farm in Kudat [31]. Hence we viewed this work as a form of promoting inclusivity by inviting those that were not involved in the gazettement process to provide their thoughts which could further enhance the current management plan of TMP. In addition, this work could serve as a post-assessment for the zoning of TMP to examine whether destructive and large-scale fishing activities are still occurring in the prohibited zones. The trend of the natural resource use could be examined by comparing the digitised maps and locations of the photographs to the maps used for the establishment of the four zones within TMP. Furthermore, the results of our work indicated that there is a shift of communities' reliance on fishing for livelihood to other activities such as ecotourism (e.g. turtle hatchery near Simpang Mengayau) and cultural tourism (e.g. rock islands that are associated the folklore in Pitas), hence meeting one of the TMP's goals to reduce the impact of overfishing.

Local ecological knowledge generated from participatory mapping and Photovoice could provide support to the assessment of ES. One approach of ES assessment is the matrix which evaluates the capacity and/or potential of a habitat to supply ES [8, 62, 63]. Results from the matrix approach can facilitate the policy making and development of management measures for targeted areas, as seen in Yangtze River Delta region, China [64] and the Italian Region E-R in the Northern Adriatic Sea [65]. The matrix approach draws information from available literature, spatial land cover data and expert evaluations to score the habitat's capacity and/or

potential to provide a specific ES typically from 0 to 5, where 0 = no relevant capacity, 1 = low relevant capacity, 2 = relevant capacity, 3 = moderate relevant capacity, 4 = high relevant capacity and 5 = very high relevant capacity [8, 63]. In a separate work related to this work here [66], the matrix approach was employed to assess the capacity of the marine habitats in TMP to supply specific ES based on data from available literature (peer-reviewed and non-peer reviewed) and expert opinion (academic researchers, local governing bodies and NGOs). Some ES, especially the cultural services (e.g. ceremonial activities), could not be scored initially due to the absence of data. Consequently, this work here was conducted to obtain local ecological knowledge to fill in the gaps in the assessment matrix. Results from the participatory mapping and Photovoice here had provided strong evidence of certain marine habitats supplying specific ES. For example, the natural rock outcrop provided cultural services where the local fishermen perform ceremonial activities; such information was not available in literature. As demonstrated here, we recommend incorporating local ecological knowledge in the assessment matrix of ES where the data for a particular ES is scarce.

## Limitations and challenges

To ensure long-term success of participatory-based projects, academic researchers should learn to adapt to the limitations and challenges besides collaborating with CBOs and NGOs that are based in the sites as equal partner [27]. Here we shared the caveats of this work as a reference for future similar projects. We did not quantify the differences and similarities between the maps by the participants and maps from literature review due to the lack of such expertise in our team. Nevertheless, this provided research avenues as such maps can be overlaid to further identify the areas that should be prioritised yet overlooked [54]. However, the outputs from such participatory mapping may not meet the "scientific" quality for technical accuracy and statistical estimation as supported by our discussion of potential human errors and biases [12]. Therefore, output from this work should be considered as supplementary information rather than the final outcome.

Our choice of snowball sampling have caused us to overlook communities from other villages and islands (e.g. Balambangan and Maliangin islands) where we do not have contacts. We also heeded to the advice from our contacts to exclude villages and islands that were inaccessible and whose communities had bad experience with previous projects by other government agencies and NGOs. Consequently, we could not expand our work to other areas in the large marine park, which caused us to obtain information that was restricted to a locality. Such exclusion may have caused us to miss certain local ecological knowledge that is restricted to these areas and hence unable to map the habitats and ES with further precision. While we have not observed nor heard from our participants, such exclusion may also lead to "community conflicts" where the excluded communities may be hostile towards the participating communities as seen in Kenya [28]. In addition to having particular communities as our participants, the snowball sampling approach may explain our small number of participants as the leaders of CBO and farm owner may have invited only those that they trusted and familiar with. Nonetheless, our sample size is larger than the sample size of a participatory mapping activity in Japan [15] and similar to the sample size of another Photovoice activity in Thailand [27]. While we acknowledge the potential bias and limited information due to our small sample size, the two approaches employed here had nonetheless complemented each other. Hence in the case where the sample size is small, we suggest combining several approaches for eliciting further information from the participants.

We faced dwindling participation in our activities, especially Photovoice, which we attributed to our engagement approach. Most of our team members were based in west Malaysia

while the activities were conducted and coordinated by one member in TMP, east Malaysia. Although the team member had been actively engaging with the participants, the physical gap between the participants in east Malaysia and other team members in west Malaysia may have played a role in reducing the participants' enthusiasm in this work. Another factor of the dwindling participation is the poor phone and internet connection at the study site, which is a common problem across the Sabah state [67, 68]. This became a challenge for us to maintain in contact with the participants via call and online messenger applications. During the pre-workshop for Photovoice, some participants could not use the Google Maps in their smartphone due to absence of mobile phone signal at the venue. Moreover, few participants' smartphones did not work well with Google Maps though they were able to obtain GPS coordinates using other mobile applications. Initially, we had asked the participants to submit their photographs and captions online via a Google form to allow us to sort the data prior to the group discussion, but the poor internet connection had hindered their effort. Therefore, the physical gap between participants and researchers along with poor phone and internet connection may have affected the participation in this work.

## Conclusion

Recognising the local communities as main stakeholder and employing participatory approaches are key to successful inclusive management of protected area [1, 13, 27]. This is particularly important when attempting to understand the ES within the protected area where the main users are local communities [1, 19]. The use of participatory approaches (i.e. participatory mapping and Photovoice) here revealed the complex social, economic and ecological dynamics associated with the ES in a newly established multiple-use marine park. Participatory mapping had allowed the ES to be mapped while Photovoice had provided in-depth qualitative information that complemented the maps. When these activities were conducted with group discussions, participants could achieve consensus pertaining to the ES and associated issues within the marine park. Though informative, output from these participatory approaches should be considered carefully for management and action plans due to potential human errors and biases. The limitations and challenges of these participatory approaches here suggested the following to be considered for future work: (i) design activities that require fewer effort from participants, (ii) presence of researchers at study site to foster positive relationship with participants, (iii) engage with isolated communities for larger spatial data, and (iv) segregate participants for group-based activity to allow equal opportunity in contributing data. Nonetheless, the participatory approaches employed here had revealed the capability of local communities to provide qualitative, visual and spatial data that are useful for ecosystem-based management of TMP that aims for inclusivity.

## Supporting information

**S1 File. Information sheet for participants of participatory mapping in Malay language.** (DOCX)

**S2 File. Information sheet for participants of Photovoice activity in Malay language.** (DOCX)

**S3 File. Consent form for participants of participatory mapping in Malay language.** (DOCX)

**S4 File. Consent form for participants of Photovoice activity in Malay language.** (DOCX)

**S5 File. Questionnaire for participants of Photovoice activity in Malay language.**
(DOCX)

**S1 Table. Questions for participatory mapping's group discussion which were asked in a sequential manner.**
(DOCX)

**S2 Table. Questions for Photovoice activity's group discussion where participants need to answer based on their chosen photograph and caption.**
(DOCX)

## Acknowledgments

The authors would like to thank the Blue Communities Research Programme for the initiation of ideas of this project, human resource support and technical guidance. The authors would like to express their gratitude to Sabah Parks (Tun Mustapha Park division) and Sofia Johari for introducing the authors to the communities in the marine park. This work is a success thanks to the support from the following community-based organisations: Kudat Turtle Conservation Society, Banggi Coral Conservation Society, *Kelab Belia Taritipan*, and *Persatuan Pelancongan Supirak*. This work was presented at The Rufford Small Grants Conference—Malaysia 2020 in January 2020.

## Author Contributions

**Conceptualization:** Voon-Ching Lim, Kamal Solhaimi Fadzil.

**Data curation:** Voon-Ching Lim.

**Formal analysis:** Voon-Ching Lim, Eva Vivian Justine.

**Funding acquisition:** Voon-Ching Lim.

**Investigation:** Voon-Ching Lim, Eva Vivian Justine, Kamila Yusof, Wan Nur Syazana Wan Mohamad Ariffin.

**Methodology:** Voon-Ching Lim, Kamal Solhaimi Fadzil.

**Project administration:** Kamila Yusof, Wan Nur Syazana Wan Mohamad Ariffin.

**Resources:** Hong Ching Goh.

**Supervision:** Hong Ching Goh.

**Visualization:** Voon-Ching Lim.

**Writing – original draft:** Voon-Ching Lim, Eva Vivian Justine, Kamila Yusof, Wan Nur Syazana Wan Mohamad Ariffin.

**Writing – review & editing:** Voon-Ching Lim, Hong Ching Goh, Kamal Solhaimi Fadzil.

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
