## [Decision Letter · Decision Letter 0]

14 Apr 2021

PONE-D-20-28597

Local knowledge of ecosystem services and perceptions of ecological, socialcultural and economic issues in Tun Mustapha Park, Malaysia

PLOS ONE

Dear Dr. Lim,

Thank you for submitting your manuscript to PLOS ONE. After careful consideration, we feel that it has merit but does not fully meet PLOS ONE’s publication criteria as it currently stands. Therefore, we invite you to submit a revised version of the manuscript that addresses the points raised during the review process.

All authors appreciate the promise of the manuscript but also raise concerns, including the need to clarify conceptual choices (e.g., around the ES framework), methodological limitations and how you minimize them, or substantive issues (e.g., around gender issues), among others.

We look forward to receiving your revised manuscript.

Kind regards,

Sergio Villamayor-Tomas

Academic Editor

PLOS ONE

Journal Requirements:

2. In your Methods section, please provide additional information about the participant recruitment method and the demographic details of your participants. Please ensure you have provided sufficient details to replicate the analyses such as: a) the recruitment date range (month and year), b) a description of any inclusion/exclusion criteria that were applied to participant recruitment, c) a table of relevant demographic details, d) a statement as to whether your sample can be considered representative of a larger population, e) a description of how participants were recruited, and f) descriptions of where participants were recruited and where the research took place.

3. We note that Figure 3 includes an image of a [patient / participant / in the study]. 

Additional Editor Comments (if provided):

All authors appreciate the promise of the manuscript but also raise concerns, including the need to clarify conceptual choices (e.g., around the ES framework), methodological limitations and how you minimize them, or substantive issues (e.g., around gender issues), among others.

Reviewers' comments:

Reviewer's Responses to Questions

**Comments to the Author**

1. Is the manuscript technically sound, and do the data support the conclusions?

Reviewer #1: No

Reviewer #2: Yes

Reviewer #3: Yes

2. Has the statistical analysis been performed appropriately and rigorously? 

Reviewer #1: I Don't Know

Reviewer #2: Yes

Reviewer #3: N/A

3. Have the authors made all data underlying the findings in their manuscript fully available?

Reviewer #1: No

Reviewer #2: Yes

Reviewer #3: Yes

4. Is the manuscript presented in an intelligible fashion and written in standard English?

Reviewer #1: No

Reviewer #2: Yes

Reviewer #3: Yes

5. Review Comments to the Author

Reviewer #1: The manuscript focuses on a newly established multiple-use marine park in Malaysia, Tun Mustapha Park (TMP), which aims for inclusive management, versus the historical top-down park management approach. This manuscript explores different participatory approaches for facilitating comprehensive understanding of the nature and dynamics of ecosystem services in the TMP. Community-based organizations and mariculture in the TMP were invite to help map the ecosystem services, highlight related issues, and provide recommendations. The authors suggest that, while the participatory approach empowered the communities to have a voice, due to potential errors and biases of participants, the output should be considered supplemental to the planning and management of TMP. They conclude that, although the participatory approach can present challenges, it enables local communities to provide crucial information for the effective management of protected areas.

While the study of a multi-use marine park is of interest, the manuscript has five major weaknesses: 1) Misspecfication of ecosystem services, especially cultural services; 2) lack of methodological effectiveness; 3) lack of adequate participant sample size and diversity; and Each of these are addressed below along with some additional comments.

Comment 1: Ecosystem services misspecification – The authors provide examples of the four categories of ecosystem services but some of these are misspecfied, especially cultural services. Throughout the manuscript the authors refer to the human values and activities associated with cultural services as the services themselves. For example, the authors exemplify cultural services with recreation, education, spiritual beliefs, cultural heritage, traditional skills, tourism, craft making, storytelling, local myths and folklores, etc. (lines 45, 78, Table 1). Per their definition on lines 42-44 (citing the Millennium Ecosystem Assessment), ecosystem services “are the benefits including goods and services that human received from the ecosystems”. They are not the activities or values associated with those services. For example, forage supply is a provisioning service, whereas livestock grazing is the human activity that takes advantage of that ecosystem service. In the case of cultural services, the presence of attractive or otherwise meaningful place of interest (such as a natural rock outcrop) is the cultural service and not the human values of activities associated with it, e.g., recreation, education, or religious activities. Additionally, the authors state that they combined regulating and supporting services because the participants did not understand the differences. These two categories are fundamentally different – supporting services underpin provisioning, regulating and cultural services and are not synonymous with regulating services. Examples of supporting services are photosynthesis, soil genesis, landscape formation, whereas regulating services are those services that maintain beneficial environmental conditions, including carbon sequestration (enabled by photosynthesis and carbon capture by various organisms, both supporting services), water filtration, nutrient cycling, etc.

Comment 2: Methodological limitations – The authors honestly and clearly list the limitations of their study that have led to potential errors and biases. These included: (i) activities that require too much effort from participants, (ii) limited presence of researchers at study site to foster positive relationship with participants, (iii) non-inclusion of isolated communities to ensure broad spatial data, and (iv) lack of separation of participants that enabled some participants to dominate in the contribution of information. Due to these limitations, the project seems to be more of a pilot study to test the research methodology than a study that provides sufficiently rigorous data to draw reliable conclusions.

Comment 3: Participant sample limitation – The authors report a total of 42 people in two groups (instead of about 60 participants that had been planned for three groups) in the participatory mapping part of the study, and only 16 people (in part due to the onset of the Coivd-19 outbreak) substantially different age groups in the Photovoice aspect of the project. The small non-inclusive participant sample results in a lack of confidence in the accuracy and representativeness of the participant input.

Comment 4: The authors describe the combined use of participatory mapping to integrate local perceptions, knowledge and values of ecosystem services in the assessment of the spatial distribution of the services in the TMP with the use of Photovoice to solicit “in depth individual knowledge” to supplement consensus derived from participatory mapping. The manuscript lacks a description how consensus was reached. Further, while the combination of participatory mapping and Photovoice documentation of ecosystem services is suitable for the project, it does not represent sufficiently innovative methodology for the manuscript to be publishable given the data limitations.

Comment 5: The manuscript is replete with grammatical errors and poor syntax and requires substantial editing.

Comment 6: In the title and throughout the manuscript, the word social-cultural should be hyphenated.

Comment 7: Although community members were engaged in developing ecosystem services maps, these is no indication how community members participate in the management of the TMP? What incentives are being created for the ecosystem services to be utilize sustainably?

Reviewer #2: In this article the authors used multiple qualitative and quantitative methods to investigate how local community members, and members of a local industry (shrimp farming) view the ecosystem services originating from a newly formed coral reef protected area. This is an important work because understanding people’s motivations can lead to drastically more efficient conservation action, thus this work should have a reach beyond Malaysia. ‘

The results from the participatory mapping showed that provisioning services including fishing, were the main ecosystem services being produced. Of course, one wonders how effective of a park this is going to be if so many people depend on fishing from within the boundaries. This is an important point that I would like to see the authors expand more upon

The photovoice results indicated that the environment, the economy and sociocultural services were viewed as major contributions from the reef. I would have liked a larger sample size, so I am tempering the impact of this complementary results. However there does seem to be some concordance among the two methods and I think photovoice is a nice qualitative meat along the “bones” of the participatory mapping.

Overall, I thought this was a nice study, although I do have some minor points below. I would however like to see the conversation expanded in two points. First is the conservation implications – how would these data be used to inform management? Is there a specifc management intervention that could address these services, especially concerning the large amount of fishing going on. The second, is that I would like to see more thought given to the gender disparities. What data within the mapping may be biased because of a largely male view? And are the differences between the results of the two techniques due to differences in technique or differences in the gender of the individuals participating in those techniques.

Specific edits below:

Line 41: I’m not sure I agree with your definition of Ecosystem Services, the Millennium Assessment defines them as “Ecosystem services are the benefits people obtain from ecosystems, which the MA describes as provisioning, regulating, sup- porting, and cultural services.” https://www.millenniumassessment.org/documents/document.765.aspx.pdf Section 1.2

Line 103 marine protected areaS not area

Line 159 how were the initial participants selected? How might your selection influence the voices that were privileged vs. those that were not?

Table 2: The gender dynamics are quite striking with a large male dominance in the participatory mapping, while photovoice seemed to be more equitable if not female dominated. How might this influence your results?

Line 245, what does the mnemonic PHOTO stand for?

Line 426 Given the large number of community members who making a living from extractive activities from the reef, what exactly is banned from the park?

Line 434 “Small amount” or “small fee”

Line 456 here you use age as an example for differences between ESs pr, but again, might there be gendered differences as well?

Line 544 given that much of the data collected for photovoice was near the village, do you think you needed all two weeks for participants? What was the distribution of when the photos were taken? By that I mean did most participants take them right after you assigned the task, or with a day or two before the deadline? Did they take them throughout the period of time? Just trying to get a handle on if other researchers were going to base their research off of your work, what your suggestion for best practices might be.

Reviewer #3: 1. The article offers an interesting perspective to collecting community based data. The focus in on three participatory approaches, but the distinction between participatory mapping and group discussion is unclear. To me it appears to be a single approach as participatory mapping can not happen without group discussion. Also please explain why you conducted two rounds of the participatory mapping and what happened during these two rounds ? Were all participants different?

2. The article used snow-ball sampling to recruit participants, but perhaps the authors should have established a criteria to select equal number of male and female participants. Since there was no set criteria, it could prove useful to explain why few female participants were involved in the study or explain why a majority participants happened to be males. The sample size also appears to be quite small - and should be justified.

3. Authors should also provide community feedback on the digitized map - for instance was the digitized map accepted by the participants? Were there any aspects of the digitized map that was controversial or not acknowledged properly.

4. Please explain clearly how were the participants for photovoice selected? Justify your reasoning.

5. The authors should perhaps elaborate on effective mechanisms that will ease the communication channels between local municipal and communities

Overall, the paper does support literature that emphasize the voice of locals in future conservation and management efforts.

6. PLOS authors have the option to publish the peer review history of their article (what does this mean?). If published, this will include your full peer review and any attached files.

Reviewer #1: No

Reviewer #2: No

Reviewer #3: No

---

## [Author Response · Author response to Decision Letter 0]

17 May 2021

Dear editor and reviewers,

Thank you for your constructive feedback. We have attempted to address your concerns in the newly revised manuscript. Due to the limitations by the text box, some of the information may be lost. Therefore, we have also attached a rebuttal letter in the submission. For individual responses, please see our response in blue and the revised sentences in the manuscript in red. We hope that our attempt of addressing your concerns and our newly revised manuscript could meet your expectation. We have also attached the revised figures for Figure 2 and 4, as well as the consent form from the participant for their photos to be published as Figure 3. 

Once again, thank you for your time and feedback.

Thank you,

Voon-Ching Lim

School of Science

Monash University Malaysia

-------------

Editor

Please ensure that your manuscript meets PLOS ONE's style requirements, including those for file naming. The PLOS ONE style templates can be found at https://journals.plos.org/plosone/s/file?id=wjVg/PLOSOne_formatting_sample_main_body.pdf and https://journals.plos.org/plosone/s/file?id=ba62/PLOSOne_formatting_sample_title_authors_affiliations.pdf

We have checked the format of our revised draft and author affiliation which adheres to the given guideline.

In your Methods section, please provide additional information about the participant recruitment method and the demographic details of your participants. Please ensure you have provided sufficient details to replicate the analyses such as: a) the recruitment date range (month and year), b) a description of any inclusion/exclusion criteria that were applied to participant recruitment, c) a table of relevant demographic details, d) a statement as to whether your sample can be considered representative of a larger population, e) a description of how participants were recruited, and f) descriptions of where participants were recruited and where the research took place.

Thank you for the suggestion. We have included the requested information. Please see our response to the reviewer’s comments below. 

We note that Figure 3 includes an image of a [patient / participant / in the study]. 

All authors appreciate the promise of the manuscript but also raise concerns, including the need to clarify conceptual choices (e.g., around the ES framework), methodological limitations and how you minimize them, or substantive issues (e.g., around gender issues), among others.

We have obtained the consent of the said individual for using his photograph as part of this publication. Kindly find the attached consent form that has been endorsed by the said individual and the second author.

 

Reviewer 1

The manuscript focuses on a newly established multiple-use marine park in Malaysia, Tun Mustapha Park (TMP), which aims for inclusive management, versus the historical top-down park management approach. This manuscript explores different participatory approaches for facilitating comprehensive understanding of the nature and dynamics of ecosystem services in the TMP. Community-based organizations and mariculture in the TMP were invite to help map the ecosystem services, highlight related issues, and provide recommendations. The authors suggest that, while the participatory approach empowered the communities to have a voice, due to potential errors and biases of participants, the output should be considered supplemental to the planning and management of TMP. They conclude that, although the participatory approach can present challenges, it enables local communities to provide crucial information for the effective management of protected areas.

While the study of a multi-use marine park is of interest, the manuscript has five major weaknesses: 1) Misspecfication of ecosystem services, especially cultural services; 2) lack of methodological effectiveness; 3) lack of adequate participant sample size and diversity; and Each of these are addressed below along with some additional comments.

Comment 1: Ecosystem services misspecification – The authors provide examples of the four categories of ecosystem services but some of these are misspecfied, especially cultural services. Throughout the manuscript the authors refer to the human values and activities associated with cultural services as the services themselves. For example, the authors exemplify cultural services with recreation, education, spiritual beliefs, cultural heritage, traditional skills, tourism, craft making, storytelling, local myths and folklores, etc. (lines 45, 78, Table 1). Per their definition on lines 42-44 (citing the Millennium Ecosystem Assessment), ecosystem services “are the benefits including goods and services that human received from the ecosystems”. They are not the activities or values associated with those services. For example, forage supply is a provisioning service, whereas livestock grazing is the human activity that takes advantage of that ecosystem service. In the case of cultural services, the presence of attractive or otherwise meaningful place of interest (such as a natural rock outcrop) is the cultural service and not the human values of activities associated with it, e.g., recreation, education, or religious activities. Additionally, the authors state that they combined regulating and supporting services because the participants did not understand the differences. These two categories are fundamentally different – supporting services underpin provisioning, regulating and cultural services and are not synonymous with regulating services. Examples of supporting services are photosynthesis, soil genesis, landscape formation, whereas regulating services are those services that maintain beneficial environmental conditions, including carbon sequestration (enabled by photosynthesis and carbon capture by various organisms, both supporting services), water filtration, nutrient cycling, etc.

Thank you for highlighting this. We have redefined our definition and categories of ecosystem services following Millennium Ecosystem Assessment (2003) and TEEB (2010). 

In regards to cultural services, Millennium Ecosystem Assessment (2003) categorised education under cultural services (see Figure 1.1 and Box 1.2) while TEEB (2010, pg. 37) defined cultural services as:

“Aesthetic appreciation and inspiration for culture, art and design: Language, knowledge and appreciation of the natural environment have been intimately related throughout human history.” 

While education is an activity rather a form of service, the ecosystem provides opportunities for improving human knowledge. Likewise, the ecosystem provides inspiration for local and spiritual beliefs. Hence we considered the examples that we have listed in Table 1 to be appropriate for cultural services.

We have redefined our definition and categories of ecosystem services following Millennium Ecosystem Assessment (2003) and TEEB (2010). 

The changes are in lines 42-47, Table 1, Table 3 and Figure 4. 

Ecosystem services (ES) are the benefits that human obtain directly and indirectly from the ecosystems which contribute to their well-being [1-3]. Millennium Ecosystem Assessment [2] and TEEB [3] categorised ES into provisioning services (e.g. food and medicine), cultural services (e.g. recreation, spiritual experience and educational opportunities), regulating services (e.g. pollination and carbon storage), and supporting services (e.g. production of oxygen and habitats for species).

Table 1. Categories of ecosystem services used here that are related to the marine ecosystem in TMP. Categories, definitions and examples of the ecosystem services were introduced to participants before the start of activities to ensure they could provide information suitable for this study. The definitions and examples follow the glossary provided by TEEB [3].

Category of ecosytem service Definition Example of benefits from ecosystem service

Provisioning Material derived from nature, animals and plants for human use and habitats used for agricultural and aquaculture purposes Food, beverage, medicine seeds for agriculture and aquaculture 

Cultural Non-material benefits derived from nature for spiritual and mental well-being in individuals as well as beliefs and knowledge within communities. The knowledge includes educational opportunities [2]. Recreation, tourism, craft making, inspiration for culture including local myths, folklores, activities and skills pertaining local customs, traditions and religion, community-based spiritual experiences and belonging 

Regulating Natural processes that regulates and maintain the environment which benefits human Climate regulation, sea erosion prevention, carbon storage, pollination, and treatment of toxic elements 

Supporting Natural processes that underpins almost all services (mentioned above), specifically in maintaing the habitats for supporting the population and diversity of flora and fauna Maintenance of genetic diversity, and grounds for mating, nesting and feeding for charismatic and commercial organisms

 

Table 3. Ecosystem services depicted by the photographs, accompanying captions and responses to questions by researchers during group discussion as resulted from the Photovoice activity. Note that the information for each ecosystem services provided by the participants should be considered as the benefits they received from the ecosystem services.

Benefits from ecosystem services Associated habitat Perceived threat

Provisioning 

Source of protein (i.e. fish, crab, prawn, shellfish, cuttlefish) for subsistence and sale Shore

Mangrove

Coral reefs

Sea in general Litter

Clearing of mangroves

Riverine mangroves drying up during dry season

Illegal and unregulated fishing activities

Fish bombing

Fuel Mangrove Unregulated cutting and pollution

Material for arts and crafts (i.e. wood stump) Shore 

Material for construction (i.e. pillar) Mangrove 

Cultural 

Recreation and ecotourism Coral bar

Shore

Rock islands

Sea in general Fish bombing

Erosion

Litter

Creative activities (i.e. photography) Shore Litter

Education and research Mangrove Clearing of mangroves

Traditional knowledge, myth and belief Rock islands Litter

Landscape appreciation (i.e. sunset and unique shape of wood stumps) Shore Fish bombing

Regulating 

Prevention of sea erosion Coral bar

Mangrove Fish bombing

Regulation of air quality and surrounding temperature Mangrove Unregulated cutting and pollution

Supporting 

Shelter and nesting ground for wildlife Rock island

Mangrove

Sea in general Erosion

Riverine mangroves drying up during dry season

Illegal and unregulated fishing

 

Fig 4. Digitised maps resulted from participatory mapping showing marine-associated habitats, ecosystem services and perceived threats in Tun Mustapha Park. (a) provisioning services. (b) cultural services. (c) regulating and supporting services. (d) perceived threats and pressures to the habitats and services. The information for each category of ecosystem service provided by the participants is considered as the benefits obtained from the marine ecosystem in TMP. The map was magnified to show the details. Note that there is a small island at the south of Maliangin Island called Maliangin Kecil Island which is surrounded by corals and seagrasses according to participants. LRFT stands for live reef fish trade.

We have also included a statement that the information provided by the participants should be referred as benefits of the ecosystem services in line 165-169 and Table 3.

Following the definition and categories of ES by Millennium Ecosystem Assessment [2] and TEEB [3], we considered the information provided by the participants in regards to human activities (e.g. recreation), goods (e.g. seafood) and services (e.g. nesting and feeding grounds for commercial species) in a specific habitat as benefits they received from ES. 

References:

Millennium Ecosystem Assessment. (2003). Ecosystems and Human Well-being: A Framework for Assessment. Washington, DC: Island Press.

TEEB. (2010). The Economics of Ecosystems and Biodiversity: Mainstreaming the Economics of Nature: A synthesis. Malta: Progress Press.

Comment 2: Methodological limitations – The authors honestly and clearly list the limitations of their study that have led to potential errors and biases. These included: (i) activities that require too much effort from participants, (ii) limited presence of researchers at study site to foster positive relationship with participants, (iii) non-inclusion of isolated communities to ensure broad spatial data, and (iv) lack of separation of participants that enabled some participants to dominate in the contribution of information. Due to these limitations, the project seems to be more of a pilot study to test the research methodology than a study that provides sufficiently rigorous data to draw reliable conclusions.

Thank you for the constructive feedback. We have now rephrased our title to reflect our work as a case study of using participatory mapping and Photovoice to elicit local knowledge of ecosystem services.

Eliciting local knowledge of ecosystem services using participatory mapping and Photovoice: A case study of Tun Mustapha Park, Malaysia

While this work here faced several limitations and challenges, we are confident that it could still support to the management of TMP. Furthermore, we would like to inform that this work had provided support to another separate work that attempted to assess ecosystem services across case studies in Malaysia, Indonesia, the Philippines and Vietnam using the matrix approach. The information is available in line 710-760.

The main driver of the gazettement of TMP is the unsustainable fishing activities, with the aim of minimising the impacts of fishing activities through establishment of four zones to cater different scales of activities [31-32, 35]. To reduce the local communities’ reliance on fishing activities, the state government has been actively promoting ecotourism as an additional livelihood for the communities through homestay, tour and recreational activities [61]. This effort appeared to be well received by the communities as our participants provided information of the recreational activities and sites during both participatory mapping and Photovoice. Furthermore, our participants also requested for the digitised maps to assist them in planning and identifying potential sites and activities for ecotourism. With the maps provided to them coupled with their own photographs from the Photovoice activity, the participants could convince other members of their community to promote sustainable community-based tourism while preserving the integrity of the marine ecosystem. One example from Donsol Island, the Philippines demonstrated how the local communities protect whale sharks as their tourism asset which attracts tourists and hence boosting the local economy through food and accommodation businesses [61]. Nevertheless, this effort will require support from the local government such as providing training (e.g. sustainable ecotourism), funding and other forms of help (e.g. permits and infrastructure) to ensure the success of sustainable community-based tourism.

During the process of the TMP’s gazettement, the CBOs were involved to provide their feedbacks except for the CBO that is based in Pitas and the mariculture farm in Kudat [31]. Hence we viewed this work as a form of promoting inclusivity by inviting those that were not involved in the gazettement process to provide their thoughts which could further enhance the current management plan of TMP. In addition, this work could serve as a post-assessment for the zoning of TMP to examine whether destructive and large-scale fishing activities are still occurring in the prohibited zones. The trend of the natural resource use could be examined by comparing the digitised maps and locations of the photographs to the maps used for the establishment of the four zones within TMP. Furthermore, the results of our work indicated that there is a shift of communities’ reliance on fishing for livelihood to other activities such as ecotourism (e.g. turtle hatchery near Simpang Mengayau) and cultural tourism (e.g. rock islands that are associated the folklore in Pitas), hence meeting one of the TMP’s goals to reduce the impact of overfishing.

Local ecological knowledge generated from participatory mapping and Photovoice could provide support to the assessment of ES. One approach of ES assessment is the matrix which evaluates the capacity and/or potential of a habitat to supply ES [62-64]. Results from the matrix approach can facilitate the policy making and development of management measures for targeted areas, as seen in Yangtze River Delta region, China [65] and the Italian Region E-R in the Northern Adriatic Sea [66]. The matrix approach draws information from available literature, spatial land cover data and expert evaluations to score the habitat’s capacity and/or potential to provide a specific ES typically from 0 to 5, where 0=no relevant capacity, 1=low relevant capacity, 2=relevant capacity, 3=moderate relevant capacity, 4=high relevant capacity and 5=very high relevant capacity [63-64]. In a separate work related to this work here [67], the matrix approach was employed to assess the capacity of the marine habitats in TMP to supply specific ES based on data from available literature (peer-reviewed and non-peer reviewed) and expert opinion (academic researchers, local governing bodies and NGOs). Some ES, especially the cultural services (e.g. ceremonial activities), could not be scored initially due to the absence of data. Consequently, this work here was conducted to obtain local ecological knowledge to fill in the gaps in the assessment matrix. Results from the participatory mapping and Photovoice here had provided strong evidence of certain marine habitats supplying specific ES. For example, the natural rock outcrop provided cultural services where the local fishermen perform ceremonial activities; such information was not available in literature. As demonstrated here, we recommend incorporating local ecological knowledge in the assessment matrix of ES where the data for a particular ES is scarce.

Comment 3: Participant sample limitation – The authors report a total of 42 people in two groups (instead of about 60 participants that had been planned for three groups) in the participatory mapping part of the study, and only 16 people (in part due to the onset of the Coivd-19 outbreak) substantially different age groups in the Photovoice aspect of the project. The small non-inclusive participant sample results in a lack of confidence in the accuracy and representativeness of the participant input.

We have attempted to recruit larger sample size with balanced demographic profile by reminding the leaders of community-based organisation and mariculture farm owner. However, we failed to meet this criteria, likely due to our recruitment method of snowball sampling. This consequently had led to the imbalanced ratio of age and gender among our participants. Nevertheless, we attempted to justify the size and the demographic profile of our participants:

Line 783-791

In addition to having particular communities as our participants, the snowball sampling approach may explain our small number of participants as the leaders of CBO and farm owner may have invited only those that they trusted and familiar with. Nonetheless, our sample size is larger than the sample size of a participatory mapping activity in Japan [15] and similar to the sample size of another Photovoice activity in Thailand [27]. While we acknowledge the potential bias and limited information due to our small sample size, the two approaches employed here had nonetheless complemented each other. Hence in the case where the sample size is small, we suggest combining several approaches for eliciting further information from the participants.

We also explained other factors that led to the imbalanced demographic profile of our participants.

line 576-579

Although we have attempted to homogenise the demographic profile of the participants for the activities (i.e. inviting members of specific age and gender), male senior members of CBOs oftentimes participated in our activities usually out of curiosity and invited by the junior members.

line 544-558.

Here, there were higher number of participants, mostly male, in participatory mapping compared to fewer participants, mostly female, in Photovoice. Although we have attempted to balance the ratio of gender among the participants by reminding the leaders of CBOs and farm owner, such gender differences in the two activities may indicate cultural and social barriers. As seen in Kenya and South Africa, there are cultural expecations of women being modest and hence, women have little opportunities to voice out and involve in any decision-making within the community [1, 28]. Such cultural and social barriers may have impeded the participation of local women in our participatory mapping which is a group-based activitiy, as they may not feel comfortable in voicing their thoughts in public. Conversely, Photovoice is an individual activity which required the participants to fill in a questionnaire themselves with their photographs as reference and this may have allowed them to pen down their individual thoughts. Therefore, Photovoice may have provided the women the confidence to document their views in photographs and narratives [28]. We also found that having a group discussion after the Photovoice activity allows the female participants to exchange views and provide support to their thoughts of the photographs and captions. 

Comment 4: The authors describe the combined use of participatory mapping to integrate local perceptions, knowledge and values of ecosystem services in the assessment of the spatial distribution of the services in the TMP with the use of Photovoice to solicit “in depth individual knowledge” to supplement consensus derived from participatory mapping. The manuscript lacks a description how consensus was reached. Further, while the combination of participatory mapping and Photovoice documentation of ecosystem services is suitable for the project, it does not represent sufficiently innovative methodology for the manuscript to be publishable given the data limitations.

Thank you for highlighting this. We apologised that we did not make it clearer in the initial manuscript. The first participatory mapping was aimed to get the participants’ feedback for creating the maps. During the second participatory mapping, the digitised maps were shown to participants. They were asked if they agree with the maps and whether they have any feedback regarding the maps. We also asked the participants if they would like to elaborate further on the activities that they have located on the map. Further feedback from the participants were taken.

We have restructured the original paragraph to incorporate this information at line 238-252.

After the first round of participatory mapping, the spatial input from the participants were digitised using an open source geographic information system application, QGIS 3.3.3 [40]. When digitising the maps, the input from participants were compared to satellite imagery in Google Earth Pro (www.google.com/earth) to cross-check the location of habitats. Locations of coral reefs provided by participants were overlapped with spatial data of corals in the region sourced from UNEP-WCMC, WorldFish Centre, WRI and TNC [41]. As several activities and habitats overlapped and considering it is impossible to show all details in the map, we generalised some of the details (e.g. fishing and coral reefs distribution) on the basis of amalgamation, exaggeration and selection following Traun, Klug, & Burkhard [42]. During the second round of participatory mapping, the digitised maps were shared with participants to obtain their consensus for validating the information and subsequent analyses. The participants were asked if they agree with the maps and whether they have any feedback regarding the maps. We also asked the participants if they would like to elaborate further on the activities that they have located on the map. Further feedback from the participants were taken with no changes made to the maps.

We apologised for not stating clearly the implications of this work and how it contributes to another work in the earlier manuscript. We have now added a section in “Discussion” to elaborate the relevance of this work for management and how it supported another work (of which the main author here is also a co-author in that separate work) in line 709-760.

Implications for management

The main driver of the gazettement of TMP is the unsustainable fishing activities, with the aim of minimising the impacts of fishing activities through establishment of four zones to cater different scales of activities [31-32, 35]. To reduce the local communities’ reliance on fishing activities, the state government has been actively promoting ecotourism as an additional livelihood for the communities through homestay, tour and recreational activities [61]. This effort appeared to be well received by the communities as our participants provided information of the recreational activities and sites during both participatory mapping and Photovoice. Furthermore, our participants also requested for the digitised maps to assist them in planning and identifying potential sites and activities for ecotourism. With the maps provided to them coupled with their own photographs from the Photovoice activity, the participants could convince other members of their community to promote sustainable community-based tourism while preserving the integrity of the marine ecosystem. One example from Donsol Island, the Philippines demonstrated how the local communities protect whale sharks as their tourism asset which attracts tourists and hence boosting the local economy through food and accommodation businesses [61]. Nevertheless, this effort will require support from the local government such as providing training (e.g. sustainable ecotourism), funding and other forms of help (e.g. permits and infrastructure) to ensure the success of sustainable community-based tourism.

During the process of the TMP’s gazettement, the CBOs were involved to provide their feedbacks except for the CBO that is based in Pitas and the mariculture farm in Kudat [31]. Hence we viewed this work as a form of promoting inclusivity by inviting those that were not involved in the gazettement process to provide their thoughts which could further enhance the current management plan of TMP. In addition, this work could serve as a post-assessment for the zoning of TMP to examine whether destructive and large-scale fishing activities are still occurring in the prohibited zones. The trend of the natural resource use could be examined by comparing the digitised maps and locations of the photographs to the maps used for the establishment of the four zones within TMP. Furthermore, the results of our work indicated that there is a shift of communities’ reliance on fishing for livelihood to other activities such as ecotourism (e.g. turtle hatchery near Simpang Mengayau) and cultural tourism (e.g. rock islands that are associated the folklore in Pitas), hence meeting one of the TMP’s goals to reduce the impact of overfishing.

Local ecological knowledge generated from participatory mapping and Photovoice could provide support to the assessment of ES. One approach of ES assessment is the matrix which evaluates the capacity and/or potential of a habitat to supply ES [62-64]. Results from the matrix approach can facilitate the policy making and development of management measures for targeted areas, as seen in Yangtze River Delta region, China [65] and the Italian Region E-R in the Northern Adriatic Sea [66]. The matrix approach draws information from available literature, spatial land cover data and expert evaluations to score the habitat’s capacity and/or potential to provide a specific ES typically from 0 to 5, where 0=no relevant capacity, 1=low relevant capacity, 2=relevant capacity, 3=moderate relevant capacity, 4=high relevant capacity and 5=very high relevant capacity [63-64]. In a separate work related to this work here [67], the matrix approach was employed to assess the capacity of the marine habitats in TMP to supply specific ES based on data from available literature (peer-reviewed and non-peer reviewed) and expert opinion (academic researchers, local governing bodies and NGOs). Some ES, especially the cultural services (e.g. ceremonial activities), could not be scored initially due to the absence of data. Consequently, this work here was conducted to obtain local ecological knowledge to fill in the gaps in the assessment matrix. Results from the participatory mapping and Photovoice here had provided strong evidence of certain marine habitats supplying specific ES. For example, the natural rock outcrop provided cultural services where the local fishermen perform ceremonial activities; such information was not available in literature. As demonstrated here, we recommend incorporating local ecological knowledge in the assessment matrix of ES where the data for a particular ES is scarce.

Comment 5: The manuscript is replete with grammatical errors and poor syntax and requires substantial editing.

Thank you for highlighting this. We have attempted to improve our use of language and we hope that the revised manuscript is now more pleasant for your reading.

Comment 6: In the title and throughout the manuscript, the word social-cultural should be hyphenated.

Thank you for highlighting this. We understand that the correct term would be “sociocultural” (https://www.merriam-webster.com/dictionary/sociocultural). We have rectified this error throughout the manuscript.

Comment 7: Although community members were engaged in developing ecosystem services maps, these is no indication how community members participate in the management of the TMP? What incentives are being created for the ecosystem services to be utilize sustainably?

Currently, the communities are yet to be directly involved in the current management of the TMP. Nevertheless, the local government had engaged with them when establishing the marine park to identify the issues and potential solutions. Consequently, the local government is now promoting eco-tourism as alternative livelihood to the fishing communities at the study site to reduce their reliance on seafood for livelihood and hence to minimise the impacts of overfishing and destructive fishing practices. This effort is still in infancy stage and had been impeded by the Covid-19 pandemic. The engagement with local communities in this work also revealed that the effort by local government is well-received by the communities, albeit require some form of support from the local government. We have discussed the progress of the effort by the local government which may be helpful in framing other strategies that could promote sustainable utilisation of ES for community-based tourism in line 710-739.

The main driver of the gazettement of TMP is the unsustainable fishing activities, with the aim of minimising the impacts of fishing activities through establishment of four zones to cater different scales of activities [31-32, 35]. To reduce the local communities’ reliance on fishing activities, the state government has been actively promoting ecotourism as an additional livelihood for the communities through homestay, tour and recreational activities [61]. This effort appeared to be well received by the communities as our participants provided information of the recreational activities and sites during both participatory mapping and Photovoice. Furthermore, our participants also requested for the digitised maps to assist them in planning and identifying potential sites and activities for ecotourism. With the maps provided to them coupled with their own photographs from the Photovoice activity, the participants could convince other members of their community to promote sustainable community-based tourism while preserving the integrity of the marine ecosystem. One example from Donsol Island, the Philippines demonstrated how the local communities protect whale sharks as their tourism asset which attracts tourists and hence boosting the local economy through food and accommodation businesses [61]. Nevertheless, this effort will require support from the local government such as providing training (e.g. sustainable ecotourism), funding and other forms of help (e.g. permits and infrastructure) to ensure the success of sustainable community-based tourism.

During the process of the TMP’s gazettement, the CBOs were involved to provide their feedbacks except for the CBO that is based in Pitas and the mariculture farm in Kudat [31]. Hence we viewed this work as a form of promoting inclusivity by inviting those that were not involved in the gazettement process to provide their thoughts which could further enhance the current management plan of TMP. In addition, this work could serve as a post-assessment for the zoning of TMP to examine whether destructive and large-scale fishing activities are still occurring in the prohibited zones. The trend of the natural resource use could be examined by comparing the digitised maps and locations of the photographs to the maps used for the establishment of the four zones within TMP. Furthermore, the results of our work indicated that there is a shift of communities’ reliance on fishing for livelihood to other activities such as ecotourism (e.g. turtle hatchery near Simpang Mengayau) and cultural tourism (e.g. rock islands that are associated the folklore in Pitas), hence meeting one of the TMP’s goals to reduce the impact of overfishing.

Reviewer 2

In this article the authors used multiple qualitative and quantitative methods to investigate how local community members, and members of a local industry (shrimp farming) view the ecosystem services originating from a newly formed coral reef protected area. This is an important work because understanding people’s motivations can lead to drastically more efficient conservation action, thus this work should have a reach beyond Malaysia. ‘

The results from the participatory mapping showed that provisioning services including fishing, were the main ecosystem services being produced. Of course, one wonders how effective of a park this is going to be if so many people depend on fishing from within the boundaries. This is an important point that I would like to see the authors expand more upon

The photovoice results indicated that the environment, the economy and sociocultural services were viewed as major contributions from the reef. I would have liked a larger sample size, so I am tempering the impact of this complementary results. However there does seem to be some concordance among the two methods and I think photovoice is a nice qualitative meat along the “bones” of the participatory mapping.

Overall, I thought this was a nice study, although I do have some minor points below. 

Thank you and we appreciate your feedback. We would like to note that the sole representative of the local industry is involved in mariculture farming instead of shrimp farming.

I would however like to see the conversation expanded in two points. First is the conservation implications – how would these data be used to inform management? Is there a specifc management intervention that could address these services, especially concerning the large amount of fishing going on. 

Thank you for highlighting this. We have elaborated the implications of this work for management of TMP and assessment of ES using the matrix approach. We would like to inform that this work was conducted to provide support to another separate work that attempted to assess ES across case studies in Malaysia, Indonesia, the Philippines and Vietnam using the matrix approach. The information is available in line 710-760.

The main driver of the gazettement of TMP is the unsustainable fishing activities, with the aim of minimising the impacts of fishing activities through establishment of four zones to cater different scales of activities [31-32, 35]. To reduce the local communities’ reliance on fishing activities, the state government has been actively promoting ecotourism as an additional livelihood for the communities through homestay, tour and recreational activities [61]. This effort appeared to be well received by the communities as our participants provided information of the recreational activities and sites during both participatory mapping and Photovoice. Furthermore, our participants also requested for the digitised maps to assist them in planning and identifying potential sites and activities for ecotourism. With the maps provided to them coupled with their own photographs from the Photovoice activity, the participants could convince other members of their community to promote sustainable community-based tourism while preserving the integrity of the marine ecosystem. One example from Donsol Island, the Philippines demonstrated how the local communities protect whale sharks as their tourism asset which attracts tourists and hence boosting the local economy through food and accommodation businesses [61]. Nevertheless, this effort will require support from the local government such as providing training (e.g. sustainable ecotourism), funding and other forms of help (e.g. permits and infrastructure) to ensure the success of sustainable community-based tourism.

During the process of the TMP’s gazettement, the CBOs were involved to provide their feedbacks except for the CBO that is based in Pitas and the mariculture farm in Kudat [31]. Hence we viewed this work as a form of promoting inclusivity by inviting those that were not involved in the gazettement process to provide their thoughts which could further enhance the current management plan of TMP. In addition, this work could serve as a post-assessment for the zoning of TMP to examine whether destructive and large-scale fishing activities are still occurring in the prohibited zones. The trend of the natural resource use could be examined by comparing the digitised maps and locations of the photographs to the maps used for the establishment of the four zones within TMP. Furthermore, the results of our work indicated that there is a shift of communities’ reliance on fishing for livelihood to other activities such as ecotourism (e.g. turtle hatchery near Simpang Mengayau) and cultural tourism (e.g. rock islands that are associated the folklore in Pitas), hence meeting one of the TMP’s goals to reduce the impact of overfishing.

Local ecological knowledge generated from participatory mapping and Photovoice could provide support to the assessment of ES. One approach of ES assessment is the matrix which evaluates the capacity and/or potential of a habitat to supply ES [62-64]. Results from the matrix approach can facilitate the policy making and development of management measures for targeted areas, as seen in Yangtze River Delta region, China [65] and the Italian Region E-R in the Northern Adriatic Sea [66]. The matrix approach draws information from available literature, spatial land cover data and expert evaluations to score the habitat’s capacity and/or potential to provide a specific ES typically from 0 to 5, where 0=no relevant capacity, 1=low relevant capacity, 2=relevant capacity, 3=moderate relevant capacity, 4=high relevant capacity and 5=very high relevant capacity [63-64]. In a separate work related to this work here [67], the matrix approach was employed to assess the capacity of the marine habitats in TMP to supply specific ES based on data from available literature (peer-reviewed and non-peer reviewed) and expert opinion (academic researchers, local governing bodies and NGOs). Some ES, especially the cultural services (e.g. ceremonial activities), could not be scored initially due to the absence of data. Consequently, this work here was conducted to obtain local ecological knowledge to fill in the gaps in the assessment matrix. Results from the participatory mapping and Photovoice here had provided strong evidence of certain marine habitats supplying specific ES. For example, the natural rock outcrop provided cultural services where the local fishermen perform ceremonial activities; such information was not available in literature. As demonstrated here, we recommend incorporating local ecological knowledge in the assessment matrix of ES where the data for a particular ES is scarce.

The second, is that I would like to see more thought given to the gender disparities. What data within the mapping may be biased because of a largely male view? And are the differences between the results of the two techniques due to differences in technique or differences in the gender of the individuals participating in those techniques.

Thank you for highlighting this. We have attempted to discuss the differences in the information generated by participatory mapping and Photovoice in relation to socio-demography of the participants in line 610-630.

As discussed above, participatory mapping allows inclusion of more people to capture more collective thoughts whereas Photovoice provides in-depth individual knowledge that supplements the consensus resulted from participatory mapping [1]. The two activities generated many similar information of activities that benefit from the ES, particularly fishing and recreational activities. However, there were information which were generated from the Photovoice that were not captured by the participatory mapping, such as the regulating services. Furthermore, gendered differences were observed which can be attributed to the domination of the participatory mapping by male participants whose feedbacks focused on economic activities, while the Photovoice activity was dominated by female participants who tend to voice their environmental concerns. It is unclear whether such differences were due to the two approaches employed here or influenced by the socio-demographic factors. The participants of Photovoice were mostly young female and have received ecological knowledge from the researchers and possibly from school. Note that the participatory mapping was also dominated by older male participants who were considered as senior members of the communities. Examining the factors that lead to the differences would require further employment of the two approaches in other communities with larger sample size that has a balanced socio-demographic profile. Nonetheless, when conducting group-based activities such as participatory mapping and group discussion, participants should be segregated based on gender, age and education as well as position in a particular organisation or community to minimise the domination of discussions and hence bias in feedback as well as to capture the different social dimension of ES [12, 38; 54].

Specific edits below:

Line 41: I’m not sure I agree with your definition of Ecosystem Services, the Millennium Assessment defines them as “Ecosystem services are the benefits people obtain from ecosystems, which the MA describes as provisioning, regulating, sup- porting, and cultural services.” https://www.millenniumassessment.org/documents/document.765.aspx.pdf Section 1.2

Thank you for sharing the document. We have redefined our definition and categories of ecosystem services following Millennium Ecosystem Assessment (2003) and TEEB (2010). 

The changes are in lines 42-47 and Table 1. 

Ecosystem services (ES) are the benefits that human obtain directly and indirectly from the ecosystems which contribute to their well-being [1-3]. Millennium Ecosystem Assessment [2] and TEEB [3] categorised ES into provisioning services (e.g. food and medicine), cultural services (e.g. recreation, spiritual experience and educational opportunities), regulating services (e.g. pollination and carbon storage), and supporting services (e.g. production of oxygen and habitats for species).

Table 1. Categories of ecosystem services used here that are related to the marine ecosystem in TMP. Categories, definitions and examples of the ecosystem services were introduced to participants before the start of activities to ensure they could provide information suitable for this study. The definitions and examples follow the glossary provided by TEEB [3].

Category of ecosytem service Definition Example of benefits from ecosystem service

Provisioning Material derived from nature, animals and plants for human use and habitats used for agricultural and aquaculture purposes Food, beverage, medicine seeds for agriculture and aquaculture 

Cultural Non-material benefits derived from nature for spiritual and mental well-being in individuals as well as beliefs and knowledge within communities. The knowledge includes educational opportunities [2]. Recreation, tourism, craft making, inspiration for culture including local myths, folklores, activities and skills pertaining local customs, traditions and religion, community-based spiritual experiences and belonging 

Regulating Natural processes that regulates and maintain the environment which benefits human Climate regulation, sea erosion prevention, carbon storage, pollination, and treatment of toxic elements 

Supporting Natural processes that underpins almost all services (mentioned above), specifically in maintaing the habitats for supporting the population and diversity of flora and fauna Maintenance of genetic diversity, and grounds for mating, nesting and feeding for charismatic and commercial organisms

Subsequently, we also restructured our results and discussion in accordance to the changes to the definition in line 360-371:

Regulating services

When asked, participants considered animals only despite the questions were framed to guide them to consider abiotic components too. They mentioned that there were beehives of stingless bees near Marudu Bay (Fig 4c), mostly as apiculture by locals, which play a role in pollinating the mangrove trees near the bay besides providing honey.

Supporting services

The participants located hatchery and foraging grounds for turtles on the map (Fig 4c). The hatchery was set up by a local youth club, Kudat Turtle Conservation Society (KTCS), whereas the locations of feeding ground were based on their observation during fishing. According to the participants, dugongs (Dugong dugon) have been sighted feeding at the south of Banggi Island. They added that these areas are important for supporting the population of these animals.

References:

Millennium Ecosystem Assessment. (2003). Ecosystems and Human Well-being: A Framework for Assessment. Washington, DC: Island Press.

TEEB. (2010). The Economics of Ecosystems and Biodiversity: Mainstreaming the Economics of Nature: A synthesis. Malta: Progress Press.

Line 103 marine protected areaS not area

We have rectified this in line 103-106.

Given the potential of participatory research to support management of marine protected areas, we employed participatory mapping and Photovoice to better understand the marine-associated habitats and ES in TMP.

Line 159 how were the initial participants selected? How might your selection influence the voices that were privileged vs. those that were not?

We elaborated details of our initial contacts and recruitment of participants in line 176-196.

We employed snowball sampling to recruit participants, where our initial contacts introduced us to potential participants who further recruited other participants [37]. Our initial contacts were past and current members of government agency and non-governmental organisation (NGO) whom we contacted through official communication. Between August 2018 and February 2019, we were introduced to four community-based organisations (CBO) and one mariculture farm by our initial contacts (Fig 1). The CBO in Pitas and the mariculture farm in Kudat did not participate in the establishment of TMP while the others did. Members of these groups were residing and/or working in the marine habitats of our interest in TMP, and hence represent the “primary users” of the ES. We first met the leaders of the CBOs and the owner of the mariculture farm, where we introduced ourselves formally, proposed our research activities (i.e. participatory mapping and Photovoice) and explained the implications of their participation in supporting the existing management of TMP. Subsequently, the leaders and farm owner recruited their members and employees respectively to participate in both activities. Although we reminded the leaders and farm owner to balance the demographic profile of their invitees (e.g. ratio of gender and age), the participants of participatory mapping were mostly older and male whereas there were more younger and female participants in Photovoice. Socio-demographic details of the participants were available in Table 2.

Participatory mapping was conducted in 2019 with the CBOs and mariculture farm whereas Photovoice was conducted in 2020 with the CBOs only as the mariculture farm did not respond to our invitation. Note that only two of all participants who joined the Photovoice also joined the participatory mapping while the rest did not join the latter. 

We also discussed briefly the impact of our recruitment and the consequent exclusion of certain participants in line 773-786.

Our choice of snowball sampling have caused us to overlook communities from other villages and islands (e.g. Balambangan and Maliangin islands) where we do not have contacts. We also heeded to the advice from our contacts to exclude villages and islands that were inaccessible and whose communities had bad experience with previous projects by other government agencies and NGOs. Consequently, we could not expand our work to other areas in the large marine park, which caused us to obtain information that was restricted to a locality. Such exclusion may have caused us to miss certain local ecological knowledge that is restricted to these areas and hence unable to map the habitats and ES with further precision. While we have not observed nor heard from our participants, such exclusion may also lead to “community conflicts” where the excluded communities may be hostile towards the participating communities as seen in Kenya [28]. In addition to having particular communities as our participants, the snowball sampling approach may explain our small number of participants as the leaders of CBO and farm owner may have invited only those that they trusted and familiar with.

Table 2: The gender dynamics are quite striking with a large male dominance in the participatory mapping, while photovoice seemed to be more equitable if not female dominated. How might this influence your results?

In the discussion section, we mentioned that the different information provided by male and female participants suggested that they perceived ecosystem services differently and hence demonstrating their distinctive interests and needs. We also discussed possible reason of fewer female participants in participatory mapping but more in Photovoice in line 544-569.

Here, there were higher number of participants, mostly male, in participatory mapping compared to fewer participants, mostly female, in Photovoice. Although we have attempted to balance the ratio of gender among the participants by reminding the leaders of CBOs and farm owner, such gender differences in the two activities may indicate cultural and social barriers. As seen in Kenya and South Africa, there are cultural expecations of women being modest and hence, women have little opportunities to voice out and involve in any decision-making within the community [1, 28]. Such cultural and social barriers may have impeded the participation of local women in our participatory mapping which is a group-based activitiy, as they may not feel comfortable in voicing their thoughts in public. Conversely, Photovoice is an individual activity which required the participants to fill in a questionnaire themselves with their photographs as reference and this may have allowed them to pen down their individual thoughts. Therefore, Photovoice may have provided the women the confidence to document their views in photographs and narratives [28]. We also found that having a group discussion after the Photovoice activity allows the female participants to exchange views and provide support to their thoughts of the photographs and captions. 

We observed that male participants tend to provide information related to economic activities, particularly fishing and mariculture farming, during the participatory mapping. In comparison, female participants focused on environmental issues during the Photovoice activity such as plastic waste at beach as well as education and photography opportunities. This suggested that men and women have distinctive interests and needs for the ES; in this case, income versus nature appreciation [54]. Furthermore, men and women are likely to interact with different parts of the marine ecosystem due to their roles, where men fish in the sea while women perform their household duties at the coastal area [55]. Women also tend to value a wider range of ES, particularly those that socially benefit the households and communities [55]. Such gendered differences in the perception of ES indicated the importance of participation of women in decision-making when managing marine protected areas.

Line 245, what does the mnemonic PHOTO stand for?

The original mnemonic PHOTO comprised the following 5 questions (Hergenrather et al., 2009):

1. “Describe your Picture.” 

2. “What is Happening in your picture?” 

3. “Why did you take a picture Of this?” 

4. “What does this picture Tell us about your life?” 

5. “How can this picture provide Opportunities for us to improve life?”

We have included the 5 questions that represent the mnemonic PHOTO in line 279-285.

Mnemonic PHOTO comprised five reflective questions which were modified from Hergenrather et al. [43]:

1. What is the story behind your Photo?

2. What are the ecosystem services Happening in your photo? 

3. Why did you take a photo Of this?

4. What are the Threats to your life or your community in this photo?

5. How can this picture provide Opportunities for things to be better in future?

Reference:

Hergenrather, K. C., Rhodes, S. D., Cowan, C. A., Bardhoshi, G., & Pula, S. (2009). Photovoice as community-based participatory research: A qualitative review. American Journal of Health Behavior, 33(6), 686-698.

Line 426 Given the large number of community members who making a living from extractive activities from the reef, what exactly is banned from the park?

We have included the details of permitted and prohibitted activities in Tun Mustapha Park in line 128-134.

Such threats were the drivers of the establishment of TMP in 2016 which is divided into four zones; (i) no-take zone where extractive activities are prohibited, (ii) community-use zone where non-destructive small-scale and traditional fishing activities are allowed and nearby communities could take part in managing the natural resources, (ii) multiple-use zone for low impact activities including non-destructive and small-scale fishing activities as well as sustainable development activities (e.g. tourism and recreation) are allowed, and (iv) commercial fishing zone where large-scale extractive fishing practices are allowed [31, 35].

Line 434 “Small amount” or “small fee”

We have changed this sentence at line 490-492.

For a small fee, the communities offer recreational-related services (e.g. guide, kayaking, homestay experience and boat services) to domestic and international tourists.

Line 456 here you use age as an example for differences between ESs pr, but again, might there be gendered differences as well?

We discussed this briefly at line 592-593 in previous manuscript. We have now moved it to this section where we elaborated further on the gendered differences in perception of ecosystem services at line 559-569.

We observed that male participants tend to provide information related to economic activities, particularly fishing and mariculture farming, during the participatory mapping. In comparison, female participants focused on environmental issues during the Photovoice activity such as plastic waste at beach as well as education and photography opportunities. This suggested that men and women have distinctive interests and needs for the ES; in this case, income versus nature appreciation [54]. Furthermore, men and women are likely to interact with different parts of the marine ecosystem due to their roles, where men fish in the sea while women perform their household duties at the coastal area [55]. Women also tend to value a wider range of ES, particularly those that socially benefit the households and communities [55]. Such gendered differences in the perception of ES indicated the importance of participation of women in decision-making when managing marine protected areas.

Line 544 given that much of the data collected for photovoice was near the village, do you think you needed all two weeks for participants? What was the distribution of when the photos were taken? By that I mean did most participants take them right after you assigned the task, or with a day or two before the deadline? Did they take them throughout the period of time? Just trying to get a handle on if other researchers were going to base their research off of your work, what your suggestion for best practices might be.

Thank you for raising this point. We have added the requested information for the Photovoice activity and compared it with participatory mapping in terms of time and effort required for the activities at line 580-609.

One factor that encourages the participation in citizen science projects is the existence of user-friendly technical tools for data collection [22-23]. Here, Photovoice requires participants to photograph certain activities or issues with their own smartphone and write descriptive captions within fourteen days before meeting the researchers. When asked, many participants admitted that they did not take photographs immediately after meeting the researchers as they need to look for inspirational places and events to help them with the Photovoice activity. For example, a group in Kota Marudu travelled to a particular coastal mangrove during the weekend for the Photovoice activity. Furthermore, photographing ES may be difficult and could not fully fit the stories or issues to be highlighted by participants [26], though we have included group discussions and questionnaire to facilitate the participants in expressing their thoughts. Photovoice also requires continuous engagement from researchers and commitment from participants. Although we tried to engage with participants frequently, there were two male participants from the participatory mapping who expressed their interest in the Photovoice activity failed to provide any photographs nor captions with the reason given being “busy”. As fewer participants agreed to meet-up during the second meeting for Photovoice coupled with Movement Control Order in Malaysia due to Covid-19 pandemic [44], we ended the Photovoice activity after two rounds.

In contrast, participatory mapping only required participants to provide their thoughts immediately during the meeting with researchers, who also guided them in pasting the stickers and drawing the features on maps. This required lesser effort from the participants, which may explain the higher turnout of participants and more information shared by the participants for the participatory mapping. As demonstrated here, the success of participatory projects depends largely on the amount of participants’ enthusiasm and whether the approach employed is user-friendly. Therefore, we recommend participatory mapping which could attract more participants and require less effort from them. Nevertheless, Photovoice could supplement the information from participatory mapping as demonstrated here. As Photovoice activity requires commitment from participants, such approach would be suitable for promoting participation of minority groups (i.e. female in this case) in any stakeholder engagement and decision-making. However, more time would be required for Photovoice to generate the desired information as participants need to search for inspirational places and events.

 

Reviewer 3

1. The article offers an interesting perspective to collecting community based data. The focus in on three participatory approaches, but the distinction between participatory mapping and group discussion is unclear. To me it appears to be a single approach as participatory mapping can not happen without group discussion. Also please explain why you conducted two rounds of the participatory mapping and what happened during these two rounds ? Were all participants different?

Thank you for highlighting this. Considering that group discussion is part of participatory mapping and Photovoice in this case, we have decided not to distinguish group discussion from the two activities. We have rephrased the sentences in the manuscript to indicate this point clearly.

Regarding the two rounds of participatory mapping, we apologised that we did not make it clearer in the initial manuscript. The first participatory mapping was aimed to get the participants’ feedback for creating the maps. During the second participatory mapping, the digitised maps were shown to participants for feedbacks and consensus. No additional details about the maps were received from the participants, other than the elaboration of the activities that they have identified on the map. We have restructured the original paragraph to incorporate this information at line 238-252.

After the first round of participatory mapping, the spatial input from the participants were digitised using an open source geographic information system application, QGIS 3.3.3 [40]. When digitising the maps, the input from participants were compared to satellite imagery in Google Earth Pro (www.google.com/earth) to cross-check the location of habitats. Locations of coral reefs provided by participants were overlapped with spatial data of corals in the region sourced from UNEP-WCMC, WorldFish Centre, WRI and TNC [41]. As several activities and habitats overlapped and considering it is impossible to show all details in the map, we generalised some of the details (e.g. fishing and coral reefs distribution) on the basis of amalgamation, exaggeration and selection following Traun, Klug, & Burkhard [42]. During the second round of participatory mapping, the digitised maps were shared with participants to obtain their consensus for validating the information and subsequent analyses. The participants were asked if they agree with the maps and whether they have any feedback regarding the maps. We also asked the participants if they would like to elaborate further on the activities that they have located on the map. Further feedback from the participants were taken with no changes made to the maps.

2. The article used snow-ball sampling to recruit participants, but perhaps the authors should have established a criteria to select equal number of male and female participants. Since there was no set criteria, it could prove useful to explain why few female participants were involved in the study or explain why a majority participants happened to be males. The sample size also appears to be quite small - and should be justified.

We have included the details of why we have fewer female participants in Photovoice compared to participatory mapping in line 544-558.

Here, there were higher number of participants, mostly male, in participatory mapping compared to fewer participants, mostly female, in Photovoice. Although we have attempted to balance the ratio of gender among the participants by reminding the leaders of CBOs and farm owner, such gender differences in the two activities may indicate cultural and social barriers. As seen in Kenya and South Africa, there are cultural expecations of women being modest and hence, women have little opportunities to voice out and involve in any decision-making within the community [1, 28]. Such cultural and social barriers may have impeded the participation of local women in our participatory mapping which is a group-based activitiy, as they may not feel comfortable in voicing their thoughts in public. Conversely, Photovoice is an individual activity which required the participants to fill in a questionnaire themselves with their photographs as reference and this may have allowed them to pen down their individual thoughts. Therefore, Photovoice may have provided the women the confidence to document their views in photographs and narratives [28]. We also found that having a group discussion after the Photovoice activity allows the female participants to exchange views and provide support to their thoughts of the photographs and captions. 

We have also justified the size of our participants in line 783-791

In addition to having particular communities as our participants, the snowball sampling approach may explain our small number of participants as the leaders of CBO and farm owner may have invited only those that they trusted and familiar with. Nonetheless, our sample size is larger than the sample size of a participatory mapping activity in Japan [15] and similar to the sample size of another Photovoice activity in Thailand [27]. While we acknowledge the potential bias and limited information due to our small sample size, the two approaches employed here had nonetheless complemented each other. Hence in the case where the sample size is small, we suggest combining several approaches for eliciting further information from the participants.

3. Authors should also provide community feedback on the digitized map - for instance was the digitized map accepted by the participants? Were there any aspects of the digitized map that was controversial or not acknowledged properly.

Thank you for raising this, allowing us to clarify our method. We have incorporated this information in line 292-298.

When the digitised maps from the first participatory mapping were shown to the participants during the second round, all of the participants agreed with the details on maps and provided only elaboration of the activities that they have located on the maps such as the details of a folklore associated with the rock islands. No changes were made to the maps after the second round of participatory mapping. The participants also asked for a copy of the maps to assist them in understanding the marine-associated habitats near their village and identifying potential sites for ecotourism.

4. Please explain clearly how were the participants for photovoice selected? Justify your reasoning.

For both participatory mapping and Photovoice, the participants were members of CBOs and employees of mariculture farm. They were invited to join the activities by their leaders and empoyer. We have further elaborated the recruitment of the participants for Photovoice in line 176-196.

We employed snowball sampling to recruit participants, where our initial contacts introduced us to potential participants who further recruited other participants [37]. Our initial contacts were past and current members of government agency and non-governmental organisation (NGO) whom we contacted through official communication. Between August 2018 and February 2019, we were introduced to four community-based organisations (CBO) and one mariculture farm by our initial contacts (Fig 1). The CBO in Pitas and the mariculture farm in Kudat did not participate in the establishment of TMP while the others did. Members of these groups were residing and/or working in the marine habitats of our interest in TMP, and hence represent the “primary users” of the ES. We first met the leaders of the CBOs and the owner of the mariculture farm, where we introduced ourselves formally, proposed our research activities (i.e. participatory mapping and Photovoice) and explained the implications of their participation in supporting the existing management of TMP. Subsequently, the leaders and farm owner recruited their members and employees respectively to participate in both activities. Although we reminded the leaders and farm owner to balance the demographic profile of their invitees (e.g. ratio of gender and age), the participants of participatory mapping were mostly older and male whereas there were more younger and female participants in Photovoice. Socio-demographic details of the participants were available in Table 2.

Participatory mapping was conducted in 2019 with the CBOs and mariculture farm whereas Photovoice was conducted in 2020 with the CBOs only as the mariculture farm did not respond to our invitation. Note that only two of all participants who joined the Photovoice also joined the participatory mapping while the rest did not join the latter. 

5. The authors should perhaps elaborate on effective mechanisms that will ease the communication channels between local municipal and communities

Thank you for highlighting this. We have added the details about the issues that were considered by the participants to be responsible for the poor waste disposal system in the island. We also recommended the local municipal to engage with the local communities to develop strategic waste management for the island. The following details are in line 642-654.

Such information are useful to policymakers and authorities involved in decision-making pertaining to the conservation of particular areas within the marine park, but requires all stakeholders including communities and authorities to communicate and collaborate [6, 12]. For example, few participants attributed the poor plaste waste disposal system and irregular waste collection in Banggi Island to the absence of a representative of the local municipal in the island and lack of communication between the local municipal with the local communities. Yet stakeholder engagement which involved local communities, local government, waste operators, industry players and academic experts is pivotal for developing strategic waste management with examples from the UK [56] and Thailand [57]. We recommend the local municipal to actively engage with the communities and right agencies through interview, questionnaire and focus group discussion to identify the issues, promote public responsibility and develop proper waste disposal system to prevent the plastic waste from further damaging the shores and affecting the well-being of the communities. 

Overall, the paper does support literature that emphasize the voice of locals in future conservation and management efforts.

Thank you and we truly appreciate it.

---

## [Decision Letter · Decision Letter 1]

14 Jun 2021

Eliciting local knowledge of ecosystem services using participatory mapping and Photovoice: A case study of Tun Mustapha Park, Malaysia

PONE-D-20-28597R1

Dear Dr. Lim,

We’re pleased to inform you that your manuscript has been judged scientifically suitable for publication and will be formally accepted for publication once it meets all outstanding technical requirements.

Kind regards,

Sergio Villamayor-Tomas

Academic Editor

PLOS ONE

Additional Editor Comments (optional):

Reviewers' comments:

Reviewer's Responses to Questions

**Comments to the Author**

1. If the authors have adequately addressed your comments raised in a previous round of review and you feel that this manuscript is now acceptable for publication, you may indicate that here to bypass the “Comments to the Author” section, enter your conflict of interest statement in the “Confidential to Editor” section, and submit your "Accept" recommendation.

Reviewer #1: All comments have been addressed

Reviewer #2: All comments have been addressed

2. Is the manuscript technically sound, and do the data support the conclusions?

Reviewer #1: Partly

Reviewer #2: Yes

3. Has the statistical analysis been performed appropriately and rigorously? 

Reviewer #1: N/A

Reviewer #2: N/A

4. Have the authors made all data underlying the findings in their manuscript fully available?

Reviewer #1: No

Reviewer #2: Yes

5. Is the manuscript presented in an intelligible fashion and written in standard English?

Reviewer #1: Yes

Reviewer #2: Yes

6. Review Comments to the Author

Reviewer #1: The authors have adequately addressed my previous comments and are now presenting it as a case study rather than as a research paper. I have no further comments.

Reviewer #2: You have addressed all of my concerns, thank you. Your reworking of the gender difference makes clear how that might be influencing the results. Additionally I appreciate the expanded discussion of what activities are permissible within an MPA. Ultimately I think this will be a useful article for others in the field.

7. PLOS authors have the option to publish the peer review history of their article (what does this mean?). If published, this will include your full peer review and any attached files.

Reviewer #1: No

Reviewer #2: No

---

## [Editor Report · Acceptance letter]

1 Jul 2021

PONE-D-20-28597R1 

Eliciting local knowledge of ecosystem services using participatory mapping and Photovoice: A case study of Tun Mustapha Park, Malaysia 

Dear Dr. Lim:

I'm pleased to inform you that your manuscript has been deemed suitable for publication in PLOS ONE. Congratulations! Your manuscript is now with our production department. 

Kind regards, 

on behalf of

Dr. Sergio Villamayor-Tomas 

Academic Editor

PLOS ONE